**Resource**

# Phosphoproteome dynamics during mitotic exit in budding yeast

Sandra A Touati[1], Meghna Kataria[1,†], Andrew W Jones[2], Ambrosius P Snijders[2] & Frank Uhlmann[1,*] (ID)

## Abstract

The cell division cycle culminates in mitosis when two daughter cells are born. As cyclin-dependent kinase (Cdk) activity reaches its peak, the anaphase-promoting complex/cyclosome (APC/C) is activated to trigger sister chromatid separation and mitotic spindle elongation, followed by spindle disassembly and cytokinesis. Degradation of mitotic cyclins and activation of Cdk-counteracting phosphatases are thought to cause protein dephosphorylation to control these sequential events. Here, we use budding yeast to analyze phosphorylation dynamics of 3,456 phosphosites on 1,101 proteins with high temporal resolution as cells progress synchronously through mitosis. This reveals that successive inactivation of S and M phase Cdks and of the mitotic kinase Polo contributes to order these dephosphorylation events. Unexpectedly, we detect as many new phosphorylation events as there are dephosphorylation events. These correlate with late mitotic kinase activation and identify numerous candidate targets of these kinases. These findings revise our view of mitotic exit and portray it as a dynamic process in which a range of mitotic kinases contribute to order both protein dephosphorylation and phosphorylation.

**Keywords** cell cycle; kinases; mitosis; phosphatases; phosphoproteomics
**Subject Categories** Cell Cycle; Post-translational Modifications, Proteolysis & Proteomics
**The EMBO Journal (2018) 37: e98745**

## Introduction

The cell cycle is a process by which cells duplicate their DNA and other cellular content during interphase and segregate their chromosomes during mitosis. This is followed by mitotic exit, culminating in cytokinesis and leading to the formation of two new daughter cells. Cell cycle events must proceed in a temporally coordinated manner to ensure faithful genome duplication and segregation. Oscillations in cyclin-dependent kinase (Cdk) activity, in complex with regulatory cyclin subunits, drive these processes. Cdk phosphorylation is opposed by Cdk-counteracting phosphatases, and

the balance of kinase and phosphatase activities is thought to define each substrate's phosphorylation status. Phosphorylation affects enzyme activity, protein interactions, and localization in various ways to bring about ordered cell cycle progression (Morgan, 2007).

In budding yeast, a single Cdk (Cdc28) binds one of nine cyclins, the identities of which depend on cell cycle stage. Over the course from G1 to mitosis, changing cyclin specificity and an overall increase in Cdk activity lead to accumulating Cdk substrate phosphorylation. Substrates reach their maximum phosphorylation at the metaphase-to-anaphase transition, after which point cyclins are degraded and Cdk activity is downregulated (Bloom & Cross, 2007). Phosphatases are thought to gain the upper hand during mitotic exit, thus leading to net substrate dephosphorylation as cells return to G1. In a quantitative model for Cdk control of the cell cycle, it has been suggested that in fission yeast, the overall level of Cdk activity, rather than cyclin identities, controls sequential substrate phosphorylation (Stern & Nurse, 1996; Coudreuse & Nurse, 2011). Equally, during budding yeast mitotic exit, the progressive quantitative decrease in Cdk activity, opposed by increasing Cdc14 phosphatase activity, has been proposed to order substrate dephosphorylation (Bouchoux & Uhlmann, 2011; Uhlmann et al, 2011).

The accurate timing of mitotic exit events is of particular importance for faithful genome inheritance. At anaphase onset, the anaphase-promoting complex/cyclosome (APC/C) is activated to catalyze securin degradation and trigger chromosome segregation. At the same time, mitotic cyclins start to be degraded, followed by Polo and Aurora kinases (Shirayama et al, 1998; Lindon & Pines, 2004; Qin et al, 2016). Condensed mitotic chromosomes must first be fully separated on an elongating anaphase spindle, before the spindle subsequently disassembles and chromosomes decondense. Only upon completion of chromosome segregation, cell division concludes with cytokinesis (Stegmeier & Amon, 2004; Sullivan & Morgan, 2007). How these mitotic exit events are controlled so that they occur in the correct order is incompletely understood.

Phosphorylation kinetics during the cell cycle are thought to be fine-tuned by kinase and phosphatase recognition motifs that are found on their substrates (Loog & Morgan, 2005; Kõivomägi et al, 2011). Cdk phosphorylates serine and threonine residues followed by a proline (S/T)P. A positive charge at the +3 position following

---

1 Chromosome Segregation Laboratory, The Francis Crick Institute, London, UK
2 Mass Spectrometry Proteomics Science Technology Platform, The Francis Crick Institute, London, UK
*Corresponding author. Tel: +44 203 796 2059; E-mail: frank.uhlmann@crick.ac.uk
†Present address: University College London Cancer Institute, London, UK

the phosphorylation site (S/T)Px(K/R) is preferred, but not essential (Ubersax *et al*, 2003; Errico *et al*, 2010; Suzuki *et al*, 2015). Additional docking sites at a distance from the phosphorylation site increase substrate recognition. The G1 cyclin Cln2 in budding yeast uses a hydrophobic patch to interact with substrate LLPP sequences, whereas the S phase cyclin Clb5 recognizes RxL motifs (Bhaduri & Pryciak, 2011; Kõivomägi *et al*, 2013; Bhaduri *et al*, 2015). Budding yeast PP2A$^{Cdc55}$ and mammalian PP2A$^{B55}$ phosphatases both show a preference for threonine over serine dephosphorylation (Cundell *et al*, 2016; Godfrey *et al*, 2017; Hein *et al*, 2017). In addition, PP2A$^{B55}$ interacts with a polybasic patch, while the PP2A$^{B56}$ isoform recognizes an LxxIxE motif (Hertz *et al*, 2016; Wang *et al*, 2016).

While Cdk provides the impetus for cell cycle progression, the oscillating phosphorylation spectrum is enriched by numerous additional kinases. Among those, the Polo and Aurora kinases are known to provide essential mitotic functions. The budding yeast Polo-like kinase Cdc5 is essential for mitotic entry, chromosome segregation, and cytokinesis (Botchkarev & Haber, 2017). The Aurora kinase Ipl1 in turn is an integral part of the mitotic chromosome biorientation and segregation mechanism, with additional functions in the control of cytokinesis in higher eukaryotes (Afonso *et al*, 2017). Polo and Aurora kinases recognize (D/E/N)x(S/T) and (K/R)(K/R)x(S/T) motifs, respectively (Mok *et al*, 2010; Alexander *et al*, 2011). Other kinases required to conclude mitosis are the budding yeast "Mitotic Exit Network (MEN)" kinases Cdc15 and Mob1-Dbf2 as well as the downstream "Regulation of Ace2 and Morphogenesis (RAM)" pathway kinase Mob2-Cbk1. Both play important roles during mitotic exit and cytokinesis. Mob1-Dbf2 and Mob2-Cbk1 belong to the "Nuclear Dbf2-related" NDR kinase family, which includes key components of the Hippo signaling pathway in higher eukaryotes. They phosphorylate serines contained in a RxxS motif, but their substrate spectrum is incompletely understood (Mah *et al*, 2005; Queralt & Uhlmann, 2008; Weiss, 2012).

Mass spectrometric phosphoproteome analysis is a powerful approach to survey the regulatory output of the cell cycle control network. We have previously analyzed protein dephosphorylation during mitotic exit following ectopic Cdc14 phosphatase expression. This has led to the identification of Cdc14 substrates required for cytokinesis. Because this analysis focused on only one regulator, Cdc14, it has inevitably underrepresented the full regulatory scope of mitotic progression (Kuilman *et al*, 2015). Others have measured substrate dephosphorylation in protein extracts obtained from mitotically arrested human cells *in vitro* or following chemical Cdk inhibition *in vivo* (McCloy *et al*, 2015; Cundell *et al*, 2016). A recent study has investigated protein phosphorylation and dephosphorylation during the fission yeast cell cycle with an emphasis on Cdk substrates (Swaffer *et al*, 2016). We now provide a comprehensive analysis, at high temporal resolution, of the phosphoproteome of budding yeast cells as they synchronously progress through mitosis. This rich kinetic representation of protein phosphorylation changes reveals sequential kinase downregulation as a contributing factor that orders protein dephosphorylation. In contrast to the perception of a period of protein dephosphorylation, we find that mitotic exit involves an equal number of proteins that gain phosphorylation. Our analysis reveals an unexpected regulatory richness of mitotic exit.

# Results and Discussion

## Phosphoproteome analysis during synchronous mitotic progression

To obtain a high-resolution view of mitotic protein phosphorylation dynamics under physiological conditions, we took advantage of genetic means to synchronize mitotic progression in budding yeast. We used cells expressing the APC/C co-activator Cdc20 under control of the galactose-inducible *GAL1* promoter. Shifting to galactose-free medium resulted in uniform cell cycle arrest at the metaphase-to-anaphase transition due to Cdc20 depletion. Cells are unable to degrade cyclins, and Cdk activity remains high in this arrest. Following galactose re-addition, cells induce Cdc20 expression and synchronously progress through the mitotic stages, anaphase, telophase, and cytokinesis before returning to G1 (Fig 1A–C). We collected cells at 5-min intervals and confirmed synchrony of mitotic progression by fluorescent activated cell sorting (FACS) analysis of DNA content, by microscopically observing spindle elongation and cytokinesis, and by monitoring the abundance and electrophoretic mobility of cell cycle markers in protein extracts that we processed for Western blotting. Cells synchronously entered anaphase 10–15 min after galactose addition, accompanied by sequential degradation of the S phase and mitotic cyclins Clb5 and Clb2, respectively. The previously characterized Cdk target Orc6, a component of the pre-replicative complex, was dephosphorylated at 20–30 min, when also the stoichiometric Cdk inhibitor Sic1 started to accumulate. Cytokinesis occurred after 30–40 min. The bulk of the above samples was processed for phosphoproteome analysis.

To quantitatively capture phosphorylation changes during mitotic progression, we employed stable isotope labeling with amino acids in cell culture (SILAC). A parallel culture was grown in medium containing heavy amino acids ($^{13}C_6$-arginine and $^{13}C_6$-lysine) and was arrested in metaphase following Cdc20 depletion. This culture served as a reference. An equally sized aliquot of the "heavy"-labeled metaphase cells was combined with each of the "light" samples throughout the timecourse of mitotic progression (Fig 1D). Following cell breakage and trypsin digestion, phosphopeptides were enriched using titanium dioxide (TiO$_2$) chromatography and analyzed using LC-MS/MS data acquisition. This allowed the quantification of 3,456 phosphosites in at least six timepoints with a localization probability score > 0.85, corresponding to 1,101 unique proteins. Imputation was used to infer phosphosite abundance at missing timepoints. Imputed values accounted for < 12% of the final dataset. Figure EV1A and B shows details of the phosphosite numbers identified at each timepoint.

## Mitotic exit comprises both protein dephosphorylation and phosphorylation

Of all identified phosphosites, 83% were phosphoserines, 15% phosphothreonine, and 2% phosphotyrosine (Fig 2A). There were no notable changes in phosphotyrosine site abundance during mitotic progression, so we did not further consider these sites in our study. We note that serine is only approximately 1.3-fold more abundant than threonine in the budding yeast proteome (Ghaemmaghami *et al*, 2003; Chong *et al*, 2015), implying that

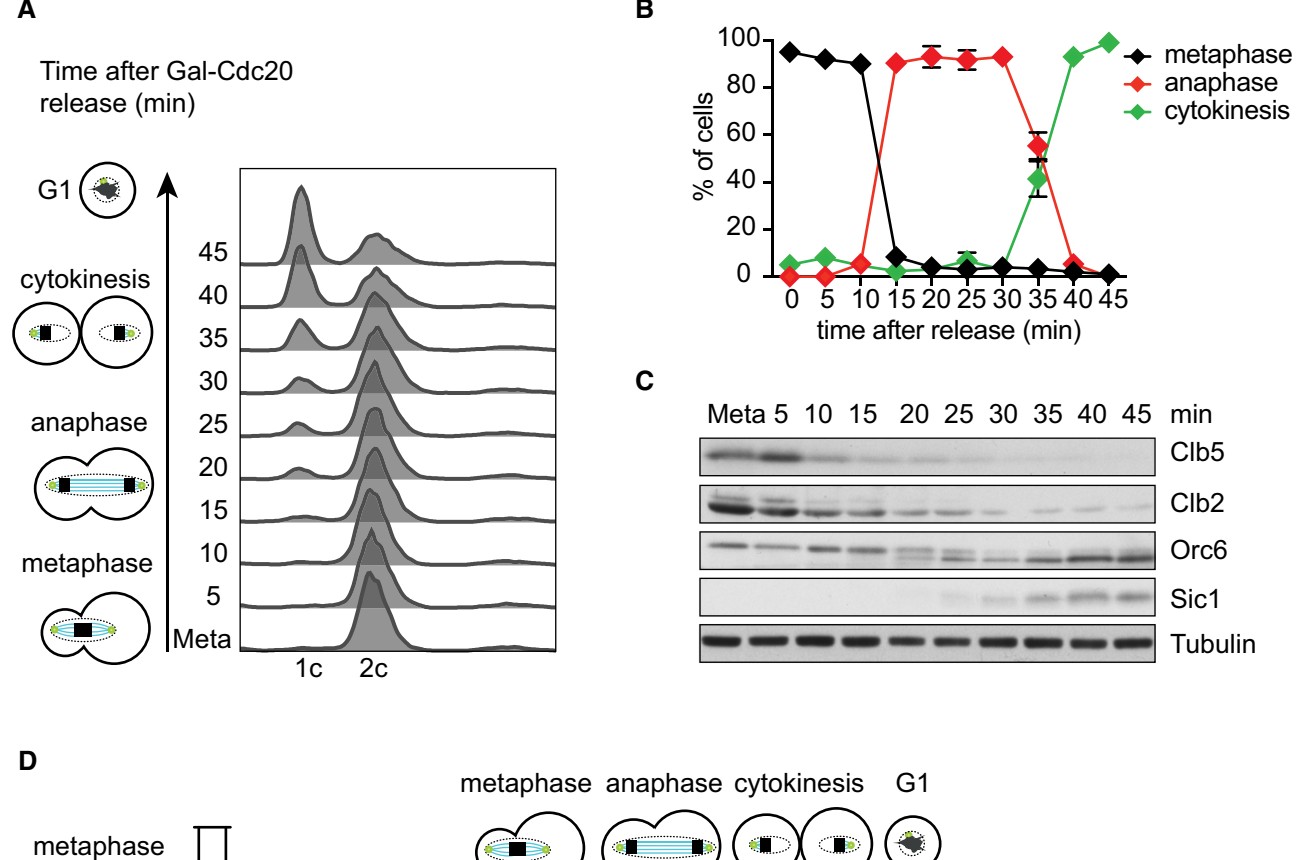

**Figure 1.  Phosphoproteome analysis during synchronized budding yeast mitotic exit.**

A   Cells were arrested in metaphase by Cdc20 depletion and then released to progress through synchronous mitosis following Cdc20 re-induction. α-factor was added to achieve eventual accumulation in G1. FACS analysis of DNA content is shown.

B   The fraction of cells with short metaphase (< 2 μm) or long anaphase (≥ 2 μm) spindles was scored in aliquots from the experiment in (A). Cytokinesis corresponds to the appearance of unbudded cells. Note that cell wall removal as part of the staining protocol results in cell separation immediately following cytokinesis, while cell separation during the FACS analysis in (A) requires new cell wall synthesis after cytokinesis. The mean ± standard deviation of three experiments is shown. One hundred cells were scored at each timepoint in each experiment.

C   Protein extracts were prepared at the indicated times and processed for Western blotting against the indicated proteins.

D   Schematic of the mass spectrometry experiment using SILAC labeling to analyze time-resolved phosphoproteome changes during mitotic progression.

Source data are available online for this figure.

serines are greatly preferred phosphoacceptor amino acids. This could be due to the greater hydrophilicity of serine, as compared to threonine, resulting in a higher probability of being solvent-exposed and accessible for modification.

We categorized phosphoserine and phosphothreonine peptides into those that were dephosphorylated, those that remained stable, and those that gained phosphorylation over the course of mitotic progression. The light-to-heavy (L/H) ratio in the first timepoint, that is, comparing cells arrested in metaphase in medium containing either light or heavy amino acids, was defined as 100%. Phosphosites whose abundance declined to < 50% over at least two consecutive timepoints were considered dephosphorylated. In turn, phosphosites whose abundance increased by more than 50% over at least two consecutive timepoints were considered phosphorylated. All other phosphosites were considered stable. Of the 3,369 phosphoserine and phosphothreonine sites, 393 (11.6%) were dephosphorylated, 339 (10.1%) were phosphorylated, while the remaining 2,637 (78.3%) stayed stable while cells progressed

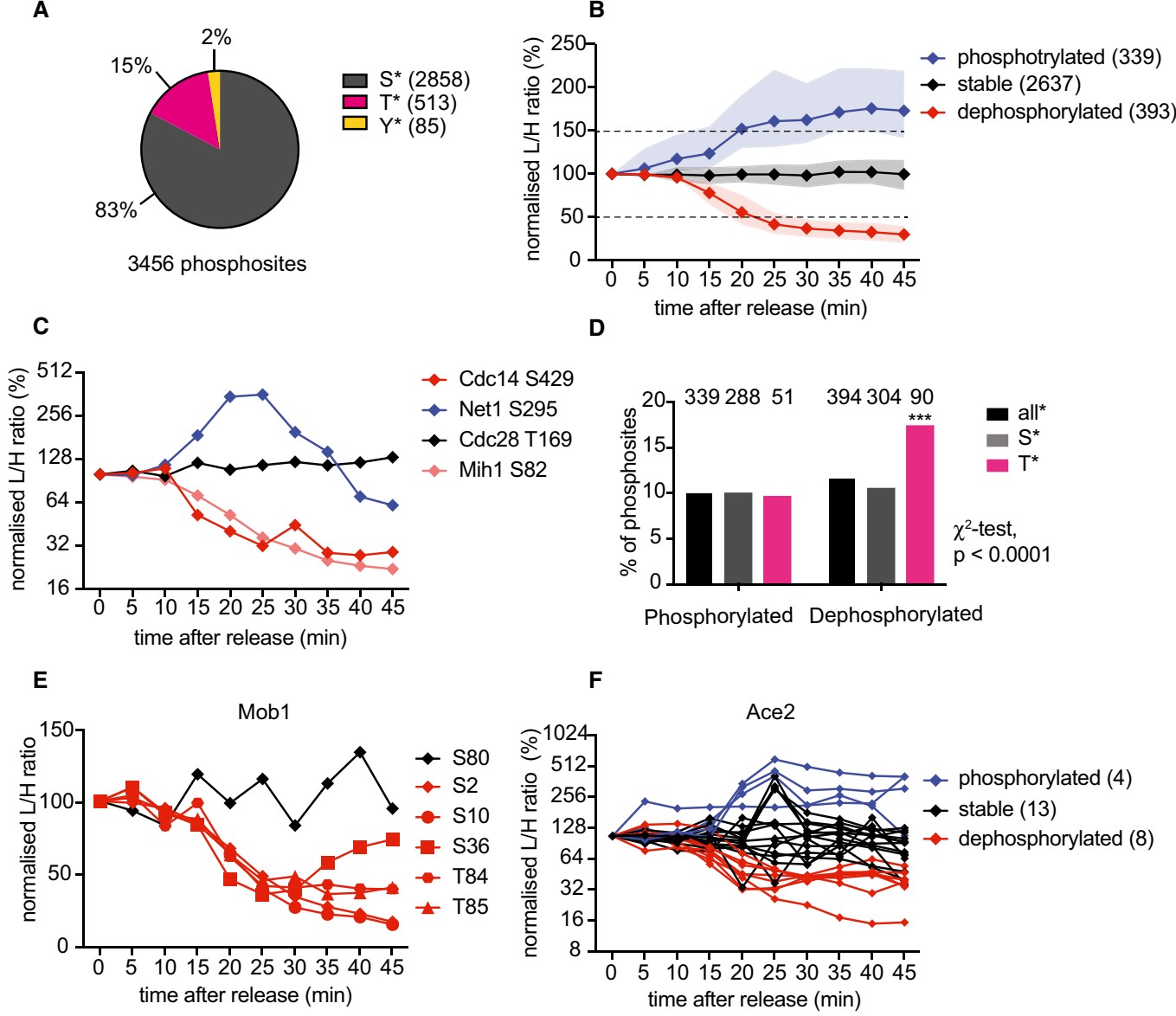

**Figure 2. Protein dephosphorylation, as well as phosphorylation, during mitotic exit.**

A   Fractions of detected serine (S), threonine (T), or tyrosine (Y) phosphosites.

B   Median intensity profiles and interquartile range of the 339 phosphosites that increased (by at least 50%), the 393 phosphosites that decreased (by at least 50%), and the 2,637 sites that remained stable during mitotic progression. See Dataset EV1 for a complete list of phosphosites in each category.

C   Profile plots of example phosphosites on known cell cycle regulators.

D   Percentages of phosphorylated and dephosphorylated sites, categorized by phosphoamino acid identity. Phosphothreonines are significantly preferred dephosphorylation targets (***$P < 0.0001$, $\chi^2$-test).

E   Profile plot of all Mob1 phosphosites. Categories stable (black) and dephosphorylated (red) are color-coded.

F   Profile plot of all Ace2 phosphosites. Categories phosphorylated (blue), stable (black,) and dephosphorylated (red) are color-coded. See Dataset EV1 for a detailed list of Ace2 phosphosites.

through and exited mitosis. Figure 2B shows the median abundance and interquartile range of the phosphosites in these three categories. A complete list of sites is contained in Dataset EV1.

In addition to phosphorylation and dephosphorylation, increased protein synthesis or protein degradation could change phosphosite abundance. In particular, the APC/C is known to target proteins for destruction during mitotic exit (Sullivan & Morgan, 2007). To gauge

the impact of protein-level changes on our analysis, we performed mass spectrometric analysis on part of our samples before phospho-peptide enrichment. This approach detected 13,209 peptides from 1,252 proteins (Fig EV1C). Based on the average of detected peptide levels for each protein, fewer than 1% were degraded (reduced to < 50% of metaphase levels), while 3% increased by more than 50% during the timecourse. Fifty-nine proteins harbor dephosphorylated

sites and were contained in both datasets, of which only one decreased in abundance. Of 74 phosphorylated proteins, two increased in abundance (Fig EV1D). Therefore, protein abundance changes do occur, but they are a relatively minor confounding factor in our phosphosite analysis.

To further validate our high-throughput phosphoproteomic analysis, we examined a set of previously characterized cell cycle regulators. Both the Cdk tyrosine phosphatase Mih1 (whose ortholog is known as Cdc25 in other organisms) and the Cdc14 phosphatase are known to be dephosphorylated during mitotic exit (Pal *et al*, 2008). Threonine 169 phosphorylation on Cdc28, the target of the Cdk-activating kinase Cak1, is thought to remain constant, while the Cdc14 inhibitor Net1 is transiently hyperphosphorylated during mitotic exit (Ross *et al*, 2000; Queralt & Uhlmann, 2008). All these expected phosphorylation changes were well recapitulated in our time-resolved dataset (Fig 2C).

We next compared the serine versus threonine representation among regulated phosphosites. Around one-tenth of all phosphoserines showed increased phosphorylation during mitotic exit. The same was observed for phosphothreonine sites, in line with the fraction of all upregulated phosphosites (Fig 2D). In contrast, 17.5% of all phosphothreonine sites were dephosphorylated during mitotic exit, a highly significant enrichment compared to 10.6% of phosphoserine sites that were dephosphorylated ($\chi^2$ test, $P < 0.0001$). This suggests that a threonine-directed phosphatase(s) makes an important contribution to budding yeast mitotic exit.

Many phosphoproteins are modified on more than one residue. Indeed, our analysis detected more than one phosphosite for over half of the recorded proteins (Fig EV1E). Often, multiple phosphosites on one protein were coordinately regulated, as seen in the example of Mob1 on which five phosphosites are lost with comparable kinetics (Fig 2E). We observed other proteins with more complex phosphorylation patterns. Figure 2F shows the example of the transcription factor Ace2. Among the 26 phosphosites detected, 4 increased, 8 decreased, and 13 remained stable during the course of mitotic progression. Ace2 accumulates in the daughter cell nucleus during mitotic exit, dependent on phosphorylation of its nuclear export sequence (NES) by the Cbk1 kinase and dephosphorylation of its nuclear localization sequence (NLS) by the Cdc14 phosphatase (Mazanka *et al*, 2008; Sbia *et al*, 2008). We found two phosphosites, S122 and S137, that are phosphorylated in the NES, as well as S714 in the NLS that was dephosphorylated (Fig EV1F). This emphasizes the importance of considering the phosphorylation dynamics of each phosphosite individually. We will analyze possible influences on individual phosphosite behavior below.

Taken together, approximately equal numbers of phosphosites are phosphorylated and dephosphorylated during progression through and exit from mitosis. This contrasts with the perception that mitotic exit is a time of protein dephosphorylation. Rather, mitotic exit emerges as a phase of dynamic changes in the phosphorylation status of a multitude of proteins.

## Kinase specificities correlate with phosphosite dephosphorylation timing

To identify possible phosphosite features that determine dephosphorylation timing, we divided the 393 dephosphorylated sites into bins

corresponding to the time when their abundance first dropped below 50%. This illustrated that protein dephosphorylation is spread out over a range of timepoints throughout mitotic progression (Fig 3A). To reveal features that correlate with dephosphorylation timing, we used IceLogo (Colaert *et al*, 2009), which uncovered enriched phosphosite characteristics in each bin. Notably, the 20-min bin was enriched for sequences that resembled the Cdk consensus phosphorylation site, while sequences in the 30-min bin resembled the Polo kinase consensus (Fig 3B). To confirm that Cdk targets are indeed dephosphorylated during mitotic progression before Polo kinase targets, we visualized all phosphosites adhering to either the full (S/T)Px(K/R) or minimal (S/T)P Cdk consensus as well as those that match the (D/E/N)x(S/T) Polo kinase motif. Plotting the distribution of 50% dephosphorylation times of these sites illustrates that Cdk consensus sites are indeed dephosphorylated distinctly earlier than Polo kinase sites (Figs 3C and EV2A). Dephosphorylation of both full and minimal Cdk consensus sites peaked 20 min after release. Full consensus sites were efficiently dephosphorylated by 25 min after release, while the dephosphorylation of minimal consensus sites stretched out for longer with substantial numbers of sites dephosphorylated as late as at the 40-min timepoint. Most Polo kinase sites were dephosphorylated between the 25- and 35-min timepoints. The same dephosphorylation order became apparent when plotting the median dephosphorylation traces and interquartile intervals (Fig EV2B).

We next studied the efficiency with which phosphosite motifs are recognized for dephosphorylation. As shown above, around one-tenth of all phosphosites in metaphase are dephosphorylated during the course of mitotic exit. This was also true when we considered those subsets that adhered to a minimal Cdk consensus motif or a Polo motif (Fig 3D). Therefore, these relatively simple motifs are not sufficient to instruct preferential substrate dephosphorylation. In striking contrast, over 60% of phosphosites that adhere to the full Cdk consensus motif (S/T)Px(K/R) were dephosphorylated. Furthermore, when we inspected dephosphorylated minimal Cdk consensus sites (i.e., (S/T)P, lacking K/R at the +3 position), we noticed that those showed an enrichment for positive charges at the +4, 5, or 6 position (Fig 3C). *In vitro* studies have shown that such positive charges enhance Cdk phosphorylation efficiency, as well as dephosphorylation by the Cdc14 phosphatase (Bremmer *et al*, 2012; Eissler *et al*, 2014; Suzuki *et al*, 2015). Our phosphoproteome survey reveals that a full Cdk consensus motif indeed bestows a particularly powerful means to convey cell cycle regulation by phosphorylation and dephosphorylation.

We have seen above that proteins often contain more than one phosphosite and that those can behave very differently during mitotic progression. One mechanism by which this could occur is that these sites adhere to different kinase recognition motifs. For example, the chitin synthase Chs2 contains phosphosites that adhere to both Cdk and Polo motifs. As expected, the Cdk sites are dephosphorylated first, followed by the Polo site (Fig 3E). In another example, the cell polarity factor Spa2 is phosphorylated on one full and six minimal Cdk consensus sites. The full consensus site is dephosphorylated more rapidly than the remaining sites (Fig 3F). Cyclin destruction during mitotic exit, especially that of the S phase cyclins, commences well before that of Polo kinase which is degraded together with the last cyclin, Clb2 (Fig EV2C), providing a possible rationale for these observations. Note that kinase motif identity must be seen as one of many factors that can

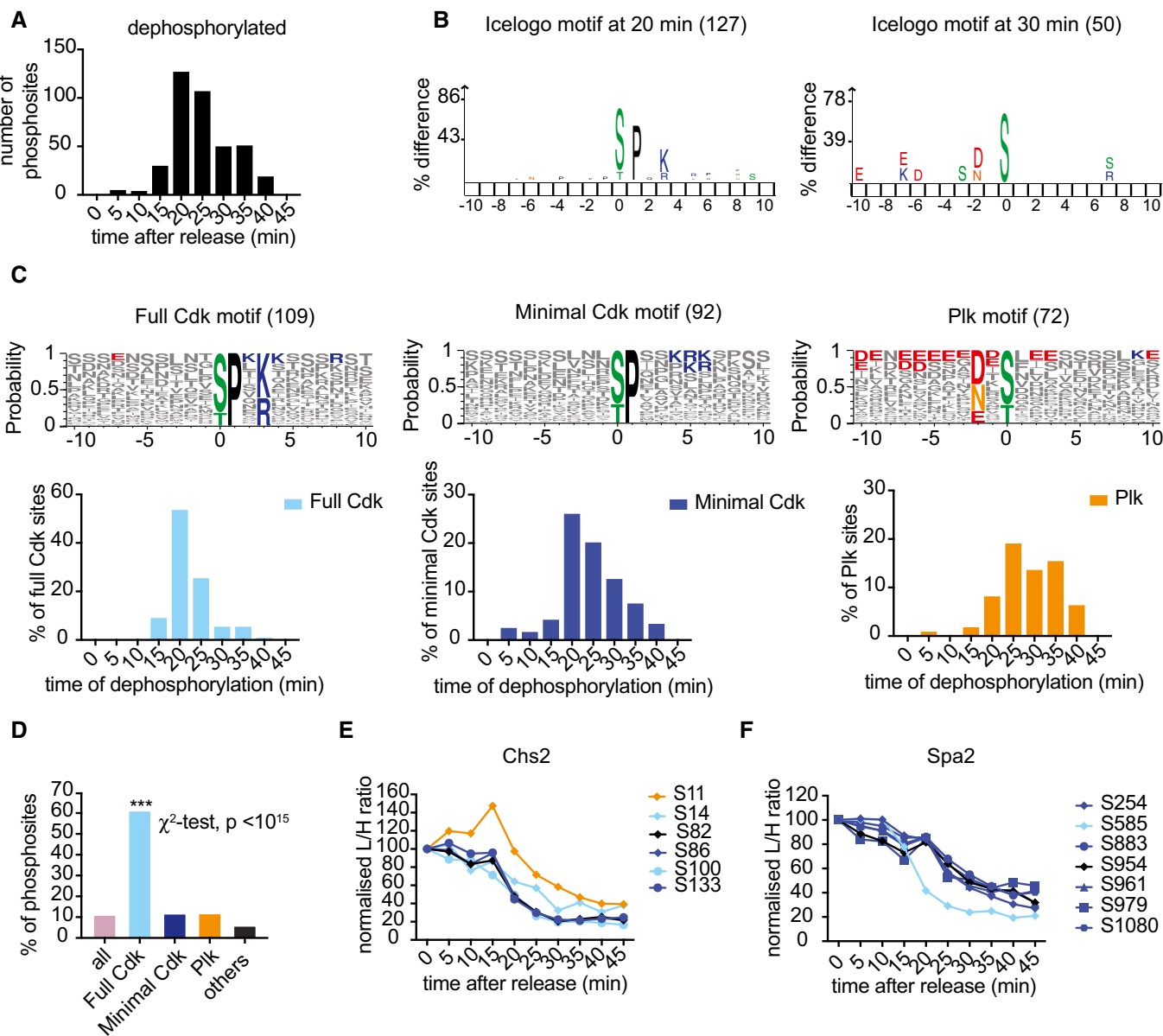

**Figure 3. Sequential Cdk and Polo site dephosphorylation.**

A  Distribution of phosphosite dephosphorylation (passing 50% threshold) timings.

B  IceLogo motif analysis of phosphosites dephosphorylated at 20 and 30 min, respectively. The phosphosite is at the center, increasing letter size indicates enrichment. The number of phosphosites dephosphorylated at both timepoints is indicated.

C  All dephosphorylated sites adhering to a full Cdk consensus motif (S/T)Px(K/R), a minimal motif (S/T)P excluding the former, and the Polo kinase motif (D/E/N)x(S/T) are aligned using Weblogo 3 to reveal additional features. The number of phosphosites in each category is indicated. Positively (blue) or negatively (red) charged amino acids are highlighted if they are the most common at any given position. Underneath, the distribution of dephosphorylation timings of sites in each category is plotted over time. See also Fig EV2A for the dephosphorylation distribution of other sites.

D  Percentages of phosphosites in each category that are dephosphorylated. Full Cdk consensus sites are significantly more likely to be dephosphorylated (***$P < 10^{-15}$, $\chi^2$-test).

E  Profile plot of Chs2 phosphosites. Phosphosite motifs are color-coded, full Cdk motif (light blue), minimal Cdk motif (dark blue), Polo motif (orange), and others (black).

F  As (E), profile plot of Spa2 phosphosites.

impact on dephosphorylation timing. For example, in the case of Chs2, most minimal and full Cdk consensus sites are dephosphorylated with similar timing, while one full Cdk consensus site is in fact dephosphorylated later (Fig 3E). Additional factors that affect

dephosphorylation timing include the specificity of phosphatases, which are as yet less well understood.

The major budding yeast mitotic exit phosphatase Cdc14 shows selectivity for phosphoserine residues (Bremmer *et al*, 2012; Eissler

*et al*, 2014). In contrast, the PP2A$^{Cdc55}$ phosphatase preferentially counteracts threonine phosphorylation (Godfrey *et al*, 2017). The human PP2A$^{Cdc55}$ ortholog, PP2A$^{B55}$, is also thought to prefer phosphothreonine substrates during mitotic exit (Cundell *et al*, 2016; Hein *et al*, 2017). We therefore explored whether serines and threonines are differentially targeted for dephosphorylation during mitotic exit. However, the average dephosphorylation timing of 90 phosphothreonine sites was indistinguishable from that of the 304 phosphoserine sites (Fig EV2D). When we specifically analyzed dephosphorylation of full Cdk consensus motifs, we found slightly later dephosphorylation of 20 TPx(K/R) sites, when compared to 89 SPx(K/R) sites (Fig EV2E). Given the small number of TPx(K/R) sites in this subset, it is not clear whether this timing difference is meaningful. We conclude that, while threonines are preferential dephosphorylation targets, their identity plays no major role in controlling the timing of their dephosphorylation during mitotic exit.

Our analysis so far was based on the in-depth technical analysis of one biological experiment. To evaluate the reproducibility of our conclusions, we performed a repeat SILAC timecourse phosphoproteomics experiment under the same experimental conditions. This dataset covered 5,723 phosphosites, including 2,931 of the 3,456 (84%) found in the first dataset. Our findings were reproducible in the repeat dataset; a summary is provided in Fig EV3.

## Clb5 degradation instructs cases of early Cdk substrate dephosphorylation

The above observations suggest that substrate dephosphorylation is in part controlled by kinases, probably by the timing of their inactivation during mitotic progression. To test this possibility experimentally, we analyzed three preferential Clb5-Cdk targets, the Yen1 resolvase, the intermediate filament protein Fin1, and a subunit of the origin recognition complex Orc6 (Loog & Morgan, 2005). We followed the electrophoretic mobility shift associated with their dephosphorylation during mitotic exit. While this is unable to resolve dephosphorylation timing of individual phosphosites, it allows us to compare approximate dephosphorylation timing between experimental conditions. To evaluate the contribution of Clb5 degradation to the dephosphorylation of these targets, we used a strain in which Clb5 was partially stabilized by deletion of its destruction box (Clb5$^{\Delta db}$; Wäsch & Cross, 2002).

Yen1 is usually an early anaphase Cdc14 substrate (Eissler *et al*, 2014). Clb5$^{\Delta db}$ protein persisted during mitotic exit and this imposed a delay to Yen1 dephosphorylation (Fig 4A). This suggests that Clb5 maintains Yen1 phosphorylation and that early Clb5 degradation in anaphase allows early Yen1 dephosphorylation. Dephosphorylation of the kinetochore protein Ask1, which is not preferentially targeted by Clb5, remained unaffected in this experiment.

Four modification sites on Fin1 were covered in our phosphoproteome analysis, including one full Cdk motif. As expected, they were dephosphorylated early, compared to average Cdk phosphosites (Fig EV4A). Clb5$^{\Delta db}$ again caused a long delay to Fin1 dephosphorylation, relative to dephosphorylation of Ask1 (Fig 4B). Even in Clb5$^{\Delta db}$ cells in G1, Fin1 was never fully dephosphorylated. To confirm that delayed Fin1 dephosphorylation is due to persistent targeting by Clb5, rather than an indirect consequence of Clb5 stabilization, we mutated the RxL motif in Fin1 that mediates Clb5

recognition (residues 194–196 "KNL" to "AAA"; Loog & Morgan, 2005). This mutation restored early Fin1 dephosphorylation in Clb5$^{\Delta db}$-expressing cells (Fig 4B), confirming that Fin1 dephosphorylation is directly controlled by Clb5. A previous study reported that Clb5$^{\Delta db}$ retains increased phosphorylation of the replication origin activator Sld2 under conditions of compromised Cdc14 activity, which further supports the notion that Clb5 degradation facilitates dephosphorylation of its substrates (Jin *et al*, 2008).

We finally analyzed Orc6, a Cdk substrate that displays exquisite Clb5 specificity (Loog & Morgan, 2005). However, its dephosphorylation timing was unaffected by Clb5 stabilization (Fig EV4B). Orc6 dephosphorylation occurs distinctly later than that of Yen1 and Fin1 (Bouchoux & Uhlmann, 2011), suggesting that in this example, the dephosphorylation timing is governed by factors other than Clb5 degradation. This emphasizes that multiple mechanisms control cell cycle phosphorylation dynamics. Phosphatase recognition, which must be considered alongside kinase specificities, might in this case exert dominant control.

## Dephosphorylation order is mirrored by rephosphorylation in the next cell cycle

We have previously reported the time-resolved phosphoproteome analysis of cells progressing synchronously from G1 to mitosis (Godfrey *et al*, 2017). This allowed us to compare the protein dephosphorylation order during mitotic exit with the order of rephosphorylation during interphase. We classified sites based on their dephosphorylation timing during mitotic exit as early (5–15 min), intermediate (20–30 min), or late (35 min or later; Figs 5A left and EV5A). Table EV1 summarizes the dephosphorylation timings of a selection of cell cycle regulators. The early, intermediate, and late dephosphorylation timings were corroborated by gene ontology (GO) analysis. Proteins harboring sites with intermediate dephosphorylation timing were significantly enriched for the GO terms mitotic cell cycle and nuclear division, while proteins harboring late dephosphorylated sites were most strongly enriched for the GO term cytokinesis (Fig EV5B).

A total of 132 of the sites dephosphorylated during mitotic exit were also contained in our interphase phosphorylation analysis and showed an at least twofold increase between G1 and mitosis (Dataset EV2; Godfrey *et al*, 2017). We now plotted the phosphorylation of these same early, intermediate, and late group members between G1 and mitosis. This revealed a striking correspondence in that early dephosphorylated sites during mitotic exit correspond to early phosphorylated sites during interphase. The same was true for intermediate and late sites (Fig 5A, right; examples of individual proteins are found in Fig EV5C).

In the search for a mechanism that explains the striking correspondence of dephosphorylation and phosphorylation timings, we turned our attention again to kinase specificity. Plotting the dephosphorylation and phosphorylation timing of Cdk and Polo consensus sites revealed that Polo sites are markedly deferred in both dephosphorylation and phosphorylation (Fig 5B). Sequential waves of kinase activity are thought to orchestrate cell cycle progression. Nevertheless, it came unexpected that waves that ebb early during mitotic exit are the same waves that rise early during the next cell cycle. Cdk activity raises early in the cell cycle and is downregulated as soon as cells enter anaphase. Polo activity in turn raises later,

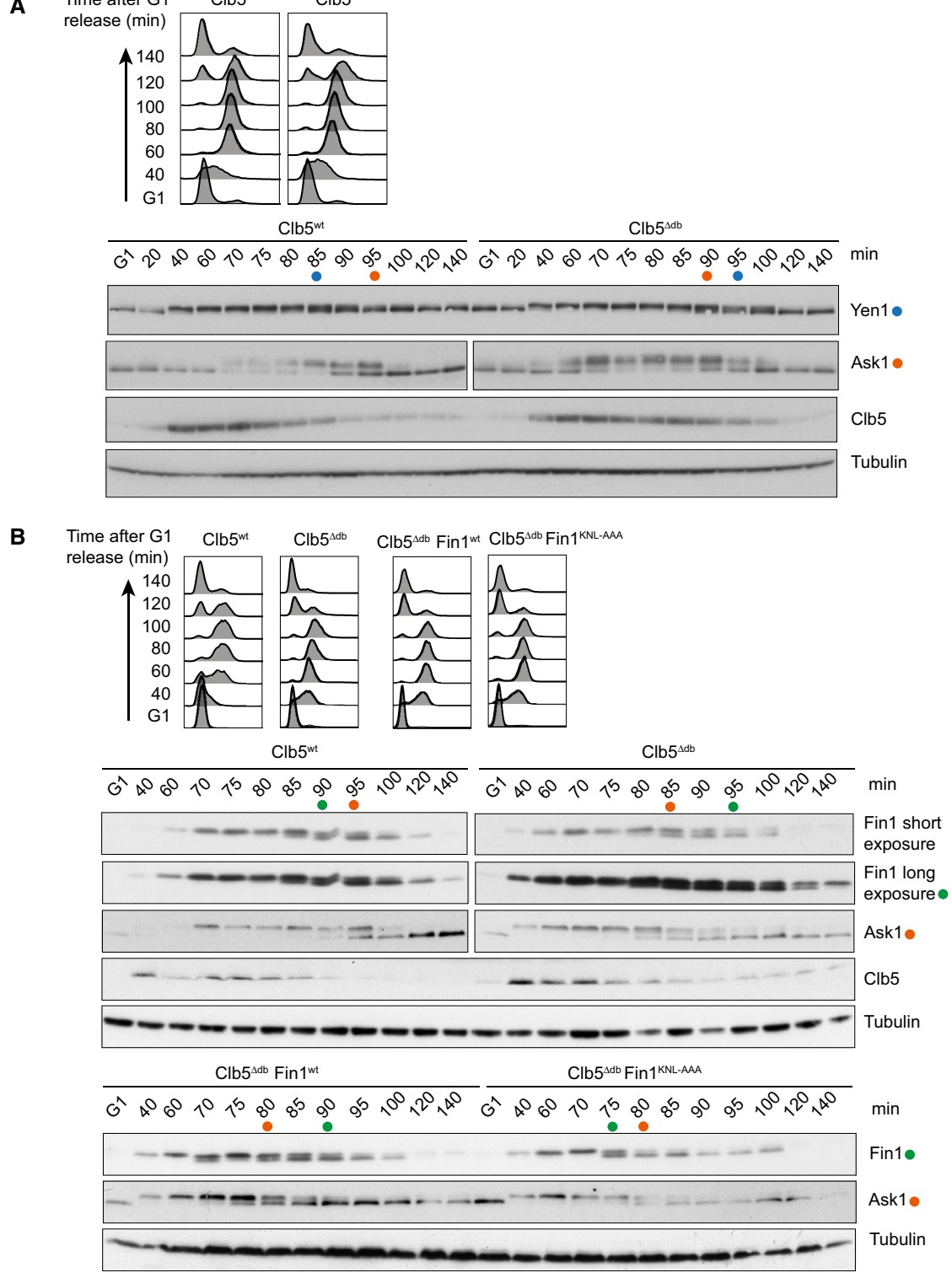

**Figure 4. Early Cbl5 degradation controls early substrate dephosphorylation.**

A   FACS analysis of DNA content confirms α-factor-induced cell cycle arrest in G1, followed by synchronous progression through one cell cycle before re-arrest in the following G1. Cells expressed wild-type Clb5 or Clb5^Δdb. Yen1 and Ask1 were fused to HA and myc epitope tags, respectively, for Western detection. Blue and orange dots indicate the relative midpoints of Yen1 and Ask1 dephosphorylation, based on their mobility shift. Clb5 levels were also analyzed. Tubulin served as a loading control.

B   As (A), but Fin1 or Fin1^KNL-AAA were fused to an HA epitope tag for detection. Green and orange dots indicate the midpoints of Fin1 and Ask1 dephosphorylation in each case.

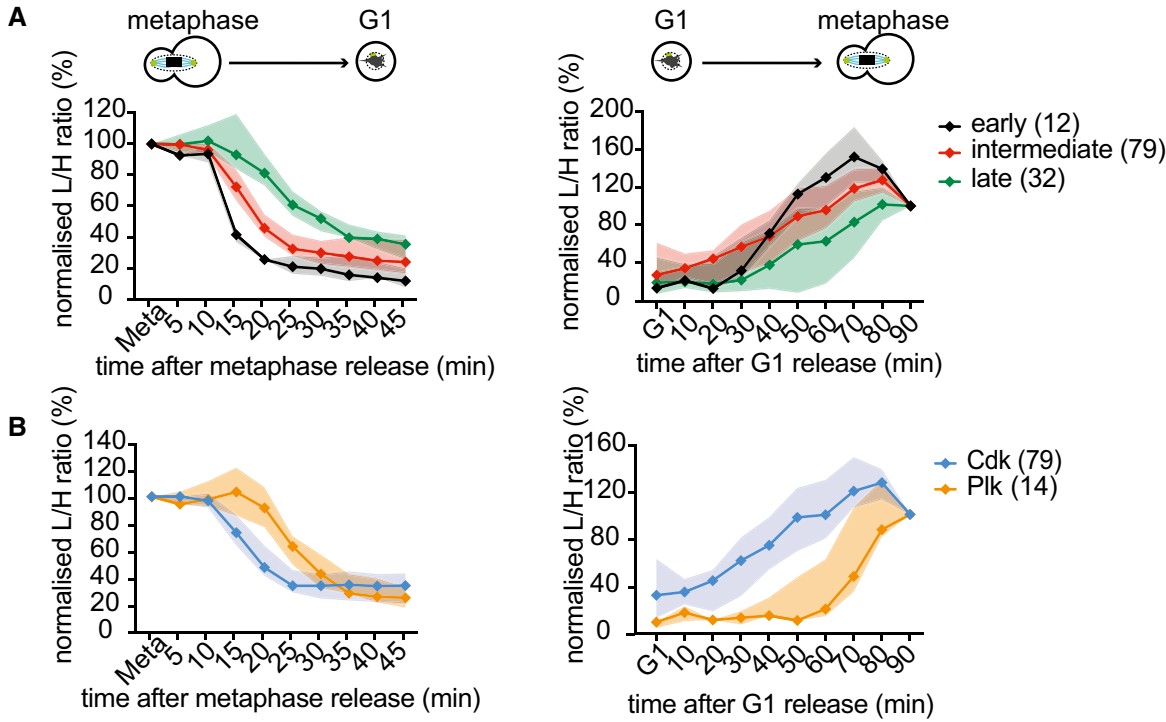

**Figure 5.  Dephosphorylation order is mirrored by phosphorylation timing during interphase.**

A  Comparison of phosphosites in common between this study and Godfrey *et al* (2017). Phosphosites were grouped by their dephosphorylation timing. The median and interquartile range of both dephosphorylation during mitotic exit as well as their phosphorylation during interphase are displayed. The number of phosphosites in each category is indicated. See also Dataset EV2 for a list of the phosphosites.

B  Median and interquartile range of Cdk (blue) and Polo (orange) phosphosites present in both studies were plotted both during mitotic exit and during interphase.

dependent on prior Cdk activation (Cheng *et al*, 1998; Mortensen *et al*, 2005), and persists for longer in mitosis. This pattern could in part be responsible for our observations. We expect that the activity windows of additional kinases, as well as phosphatases, contribute to shaping this phosphorylation landscape.

**Protein phosphorylation during mitotic exit**

Important contributions of protein kinases during mitotic exit have been documented (Afonso *et al*, 2017; Botchkarev & Haber, 2017; Gelens *et al*, 2017); however, the full extent of their action has not yet been surveyed. An important role of mitotic exit kinases in budding yeast is activation of the Cdk-counteracting phosphatase Cdc14, thereby contributing to net protein dephosphorylation. It was therefore unexpected that our phosphoproteome analysis revealed an almost equal extent of new phosphorylation during mitotic exit as there was dephosphorylation. Classifying phosphosites by the time they passed a 50% increase in abundance revealed widespread phosphorylation throughout the course of mitotic progression (Fig 6A). We again used IceLogo to interrogate phosphorylation motif enrichments at each timepoint. This revealed residual proline-directed phosphorylation at the 5-min timepoint which might be a spillover from Cdk phosphorylation in metaphase (Fig EV6A). Another small wave of (S/T)P phosphorylation started at 15 min and could be the consequence of separase-dependent PP2A^Cdc55 downregulation at this time (Fig EV6B; Queralt *et al*, 2006). Notably, the sequence logo at 15 min contained the signature

of Polo kinase, while at 20 min, the Aurora and NDR family kinase motifs dominated (Fig 6B).

We next plotted the phosphorylation time distribution of all Polo, Aurora, and NDR kinase motifs (Fig 6C). This confirmed a peak of Polo kinase site phosphorylation at 15 min, followed by peaks of Aurora kinase and the NDR kinases Dbf2 and Cbk1 at 20 min. The consensus phosphorylation site motif for Aurora and Dbf2 kinases overlaps with that of protein kinases PKA and PKB, so the latter two kinases might share responsibility for these phosphorylation events. Casein kinase 2 (CKII) motifs were also upregulated at 20 min, including the known CKII phosphosite S201 in Sic1 (Fig EV6B; Coccetti *et al*, 2004). The mean phosphosite abundance over time, separated by kinase consensus motifs, confirmed their sequential phosphorylation during mitotic exit (Fig EV6C).

We also inspected phosphorylation patterns of individual proteins displaying multiple phosphosites. Figure 6D and E shows two examples, the nucleoporin Nup60 and the Cdc14 phosphatase inhibitor Net1. Both gain phosphorylation on Polo kinase sites early during mitotic progression, followed by Aurora and Dbf2 sites. Phosphorylation on the Polo kinase sites declines again as mitotic exit progresses and Polo kinase is degraded. This illustrates how activation and inactivation dynamics of different kinases can contribute to complex phosphorylation patterns. These examples also illustrate that regulation beyond kinase control contributes to shaping the phosphorylation pattern. For example, the two Dbf2 consensus sites on Net1 show distinct phosphorylation timings. Indeed, the earlier site is phosphorylated concomitantly with a Polo consensus

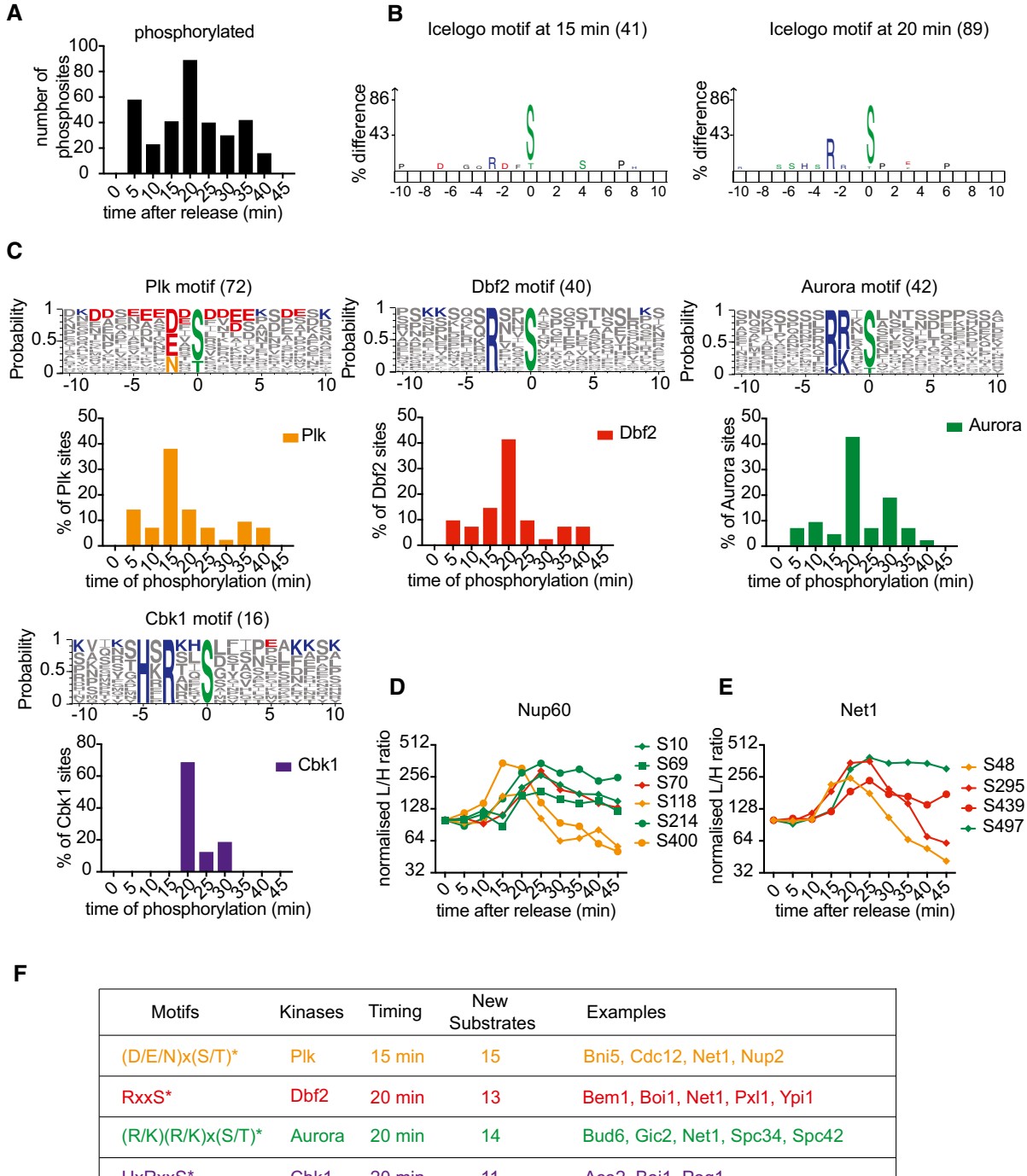

**Figure 6. Ordered substrate phosphorylation by late mitotic kinases.**

A  Distribution of phosphosite phosphorylation timings (passing 50% threshold).

B  IceLogo motif analysis of sites phosphorylated at 15 and 20 min, respectively. The phosphosite is at the center, increasing letter size indicates enrichment. The number of sites phosphorylated at both timepoints is indicated.

C  The Polo consensus motif (D/E/N)x(S/T), Dbf2 motif RxxS, Aurora motif (K/R)(K/R)x(S/T), and Cbk1 kinase motif HxRxxS were searched among all phosphorylated sites and presented using Weblogo 3 to reveal additional features. The number of phosphosites in each category is indicated. Additional positively (blue) or negatively (red) charged amino acids are highlighted if they are the most common at any given position. Underneath, the distribution of phosphorylation timings of sites in each category is plotted over time. See also Fig EV6B for the phosphorylation distribution of additional motifs.

D  Profile plot of all identified Nup60 phosphosites. Polo kinase (orange), Aurora (green), and Dbf2 motifs (red) are color-coded.

E  As (D), but Net1 phosphosites are shown.

F  Table summarizing kinase motifs, their phosphorylation timing, as well as examples of known and previously unknown substrates. See Table EV2 for a full list of substrates.

site. This emphasizes that additional mechanisms contribute to shape multisite phosphorylation patterns. For example, early phosphorylation events can prime later modifications, for example, in the case of the cytokinesis regulator Hof1 that is sequentially phosphorylated by Polo and Dbf2 kinases (Meitinger *et al*, 2011). Counteracting phosphatases add another layer of regulation.

The function of late mitotic kinases in cell physiology and cell cycle progression is only partly understood. Our analysis identified not only candidate sites but also their phosphorylation timing. Polo kinase consensus sites that are phosphorylated together with the bulk of Polo targets at 15 min can be more confidently regarded as likely Polo substrates, compared to sites that are phosphorylated at other times. Equally, NDR kinase sites phosphorylated at 20 min appear to be likely NDR targets. Combining these two criteria, phosphosite consensus motif and phosphorylation timing, led to a candidate list of previously unknown late mitotic kinase substrates (Fig 6F). These include the myosin linker Bni5 and the septin Cdc12 as Polo kinase targets and the cell polarity factors Bem1, Boi1, and Pxl1, as well as the protein phosphatase PP1 regulator Ypi1 as Dbf2 targets. We confirmed an Aurora kinase site on the kinetochore protein Spc34 and a Cbk1 kinase site on the transcription factor Ace2 (Cheeseman *et al*, 2002; Mazanka *et al*, 2008). In addition, we identified the spindle pole body component Spc42 and the cell cycle transcription factor Pog1 as potential new Aurora and Cbk1 kinase targets, respectively. A full list of candidate substrates can be found in Table EV2, which provides a resource for future studies into the role of late mitotic kinases.

## Conclusions

We have presented the time-resolved phosphoproteome analysis of budding yeast cells progressing synchronously through mitosis. This has revealed how multiple kinases impact on the complex sequential phosphorylation patterns as cells exit from mitosis. Mitotic exit was previously thought to be a time during the cell cycle when overall protein phosphorylation levels decline. Our study reveals that late mitotic kinases confer an unexpected extent of new phosphorylation events that together shape the dynamic phosphorylation changes as cells complete the process of cell division. Our approach recorded relative phosphorylation changes on a large number of proteins. In the future, it will be important to complement this with measurements of absolute phosphosite occupancies. Methods to do so have been developed (Presler *et al*, 2017), but limit the number of sites that can be interrogated. Kinase motif signatures are easily discernible, which allowed us to portray the impact of kinases on phosphoproteome dynamics. Phosphatases counteract all phosphorylation events, and steady-state phosphorylation levels are the consequence of relative kinase and phosphatase input. It will be important to characterize the contribution of phosphatases to mitotic phosphoproteome dynamics in the future.

# Materials and Methods

## Yeast strains and culture

Budding yeast strains were of W303 background. A list of the strains used in this study can be found in Table EV3. Epitope tagging of endogenous gene loci was performed using polymerase chain reaction (PCR)-based gene targeting. *GAL1-CDC20* promoter exchange was performed by gene targeting using a linearized plasmid. Fin1 gene mutagenesis was performed using the Q5 Mutagenesis Kit (New England Biolabs) on a plasmid, which was then used to modify the endogenous gene locus. The N-terminal Clb5$^{\Delta db}$ deletion was introduced at the endogenous gene locus, based on a previous report (Wäsch & Cross, 2002). All gene alterations were confirmed by DNA sequencing. Cells were grown in rich YP medium supplemented with 2% glucose (YPD) or 2% raffinose + 2% galactose (YP Raff + Gal) at 25°C. α-factor was used for cell synchronization in G1 as described (O'Reilly *et al*, 2012). Cell synchronization in metaphase and release into anaphase using *GAL1-CDC20* was as previously described (Uhlmann *et al*, 1999).

## Western blotting

Protein extracts for Western blotting were prepared following cell fixation using trichloroacetic acid, as described (Foiani *et al*, 1994), separated by SDS–polyacrylamide gel electrophoresis and transferred to PVDF membranes. Antibodies used for Western detection were α-Clb5 (Santa Cruz, sc20170), α-Clb2 (Santa Cruz, sc9071), α-Sic1 (Santa-Cruz, sc50441), α-Orc6 (clone SB49), and α-HA (clone 12CA5).

## Immunofluorescence

Indirect immunofluorescence was performed on formaldehyde-fixed cells. In brief, cells from 1 ml culture ($OD_{600}$ = 0.3) were collected by centrifugation and resuspended in 1 ml of ice cold fixation buffer (100 mM potassium phosphate pH 6.4, 0.5 mM $MgCl_2$, 3.7% formaldehyde) and fixed overnight. Cells were pelleted again and washed in 1 ml fixation buffer without formaldehyde and then in 1 ml spheroplasting buffer (100 mM potassium phosphate pH 7.4, 0.5 mM $MgCl_2$, 1.2 M sorbitol). Cells were spheroplasted in 0.2 ml spheroplasting buffer including 28 mM 2-mercaptoethanol and 20 U/μl Zymolase T-100 by incubation at 37°C for 45 min. Cells were carefully pelleted and washed in 0.5 ml spheroplasting buffer and were finally resuspended in 0.2 ml of the same buffer. A 15-well slide was coated with 0.1% Polylysine, and then, 10 μl of cell suspension was applied. After 15 min in a moist chamber, drops were aspirated and slides were incubated 3 min in a −20°C methanol bath and transferred for 10 s to a −20°C acetone bath. After acetone evaporated, cells were covered with 10 μl of blocking buffer (1% BSA in PBS) for 20 min. Spindles were then stained using an α-tubulin antibody (YOL 1/34, Bio-Rad) followed by an α-rat CY3 coupled antibody (Jackson Laboratory). Cells were counterstained with the DNA binding dye Hoechst 33258. Fluorescent images were acquired using an Axioplan 2 Imaging microscope (Zeiss) equipped with a 100× (NA = 1.45) Plan-Neofluar objective and an ORCA-ER camera (Hamamatsu). Spindles < 2 μm in length were scored as short (metaphase), while spindles that are 2 μm or longer were classified as long (anaphase). The appearance of unbudded cells with a single spindle aster was considered the products of cytokinesis. Note that the cell wall digestion that is part of the cell permeabilization protocol leads to cell separation immediately following cytokinesis.

**Fluorescent activated cell sorting analysis of DNA content**

Cells from 1 ml culture ($OD_{600}$ = 0.3) were collected by centrifugation and resuspended in 1 ml of ice cold 70% ethanol and keep for fixation on ice overnight. Cells were pelleted again and resuspended in 1 ml of RNase buffer (50 mM Tris/HCl pH 7.5, containing 0.1 mg/ml RNase A) and incubated at 37°C for 2 h. Cells were pelleted and resuspended in 0.4 ml of FACS buffer (200 mM Tris/HCl pH 7.5, 211 mM NaCl, 78 mM $MgCl_2$) containing 50 μg/ml propidium iodide. Samples were sonicated for 5 s, and 50 μl of the cell suspension was diluted in 0.5 ml 50 mM Tris/HCl pH 7.5. 10,000 cells per sample were read using a FACS Calibur cell analyzer (BD Biosciences), and the datafiles were curated using FlowJo software (FlowJo LLC). Note that cell separation and return to G1 in this FACS-based analysis await biological cell wall resolution, which follows cytokinesis with a time delay.

**Stable isotope labeling with amino acids in cell culture (SILAC)**

Cells for the "heavy" metaphase reference sample were grown for > eight generations in synthetic complete medium containing 0.1 mg/ml of ($^{13}C_6$,$^{15}N_4$)-arginine and ($^{13}C_6$,$^{15}N_2$)-lysine with raffinose + galactose as the carbon source (Gruhler *et al*, 2005) and then arrested in mitosis by filtration and shift to medium containing raffinose only. Upon arrest, 10 aliquots of the culture were harvested and resuspended in 20% trichloroacetic acid for protein fixation. In parallel, a "light" culture was grown in regular rich YP medium that was similarly arrested in metaphase by change of the carbon source to raffinose only. This culture was then released to progress through mitosis by re-addition of galactose. At the indicated times, aliquots of this culture were retrieved, resuspended in 20% trichloroacetic acid, and mixed with one of the metaphase aliquots of cells grown with heavy amino acids. Following acetone washes, cells were resuspended in lysis buffer (50 mM ammonium bicarbonate, 5 mM EDTA pH 7.5, 8 M urea) and opened by glass bead breakage. Protein extracts were cleared by centrifugation. Note that the different media composition of the "heavy" and "light" cultures will have caused certain protein phosphorylation differences at time zero.

**Sample preparation for mass spectrometry**

1 mg of protein sample was reduced with 5 mM dithiothreitol (56°C, 25 min), alkylated with 10 mM iodoacetamide (room temperature, 30 min, dark), and quenched with 7.5 M DTT. Samples were then diluted with 50 mM ammonium bicarbonate to reduce the urea concentration to < 2 M, prior to trypsin digestion (37°C, overnight). Peptides were then acidified and desalted using a $C_{18}$ SepPak Lite under vacuum and dried. To ensure complete digestion, peptides were further digested using Lys-C in 10% acetonitrile and 50 mM ammonium bicarbonate (37°C, 2 h), followed by trypsin digestion (37°C, overnight). Digested peptides were then desalted again and dried.

Phosphopeptide enrichment using titanium dioxide ($TiO_2$) was carried out as follows. Dried peptide mixtures were resuspended in 1 M glycolic acid + 80% acetonitrile + 5% TFA, sonicated (10 min), and added to $TiO_2$ beads (5:1 (w/w) beads:protein). The beads were washed using 80% acetonitrile + 1% TFA followed by 10% acetonitrile + 0.2% TFA and dried under vacuum centrifugation.

Flow-through fractions were retained for analysis of non-phosphorylated peptides. Phosphopeptides were eluted from the beads by adding 1% ammonium hydroxide followed by 5% ammonium hydroxide, and dried by vacuum centrifugation. Dried phosphopeptides were resuspended in 100 μl of 1% TFA, loaded onto a $C_{18}$ Stage Tip, and washed with 1% TFA followed by elution with 80% acetonitrile + 5% TFA. The eluted peptides were again dried under vacuum centrifugation. For analysis of non-phosphorylated peptides, the stored flow-through fractions were dried, desalted using a $C_{18}$ SepPak Lite under vacuum, and dried.

An LTQ-Orbitrap Velos coupled to an UltiMate 3000 HPLC system for online liquid chromatographic separation was used for data acquisition of all samples. Phosphopeptide mixtures were resuspended in 0.1% TFA, and triplicate runs were performed, each consisting of a 3-h gradient elution (75 μm × 50 cm $C_{18}$ column) with one MS/MS activation method per run: collision-induced dissociation (CID), multistage activation (MSA), and higher energy collision dissociation (HCD). Non-phosphopeptide mixtures were resuspended in 0.1% TFA and were analyzed over a 3-h gradient elution with CID used as the activation method.

**Mass spectrometry data analysis**

MaxQuant (version 1.3.0.5) was used for all data processing. The data was searched against a UniProt extracted *Saccharomyces cerevisiae* proteome FASTA file, amended to include common contaminants. Default MaxQuant parameters were used with the following adjustments: Phospho(STY) was added as a variable modification (for the phosphosamples only), Lys8 and Arg10 were the heavy labels, "filter-labeled amino acids" were deselected, least modified version of the peptides was used for quantification, requantify was selected with the instruction to keep low-scoring versions of identified peptides within parameter groups, and match between runs was selected. MaxQuant output files were imported into Perseus (version 1.4.0.2), and the normalized heavy-to-light (H:L) ratios were used for all subsequent analysis.

*Filtration, imputation, and smoothing*
Only phosphosites with a localization probability > 0.85 were included in the analysis. Phosphosites with H/L ratios for at least 6 out of the 10 timepoints were further analyzed. Imputation was then applied to fill missing values. These were imputed using the R package DMwR. Default parameters of the knnImputation function were used. Fewer than 12% of the values were imputed in the mitotic exit dataset. For data smoothing, an R script was applied in which the five nearest neighbors for each phosphosite were identified by Euclidian distance. The mean ratio of the five nearest neighbors' ratios was then substituted for each original ratio. Comparison of dephosphorylation with phosphorylation timings made use of a previously published dataset (ProteomeXchange: PXD004461; Godfrey *et al*, 2017). We reanalyzed this dataset as described above, using imputation to fill in missing values for phosphosites covered by at least 5 out of 10 timepoints. Only phosphosites present in both datasets were then taken into consideration for the comparison.

*Hierarchical clustering analysis*
Hierarchical clustering was performed in Perseus (Settings: cluster rows, Euclidian distance, do not presuppose K-means).

## Sequence logos

Sequence logos were created using IceLogo (Colaert *et al*, 2009) or Weblogo 3 (Crooks *et al*, 2004). Phosphoserines or phosphothreonines are placed at the central position within the sequence logo. A *P*-value = 0.01 was applied as detection threshold when using IceLogo. Percentage difference reflects the frequency of an amino acid in our dataset, at a given position, compared to its frequency in a reference dataset (the whole *S. cerevisiae* proteome). The height of the amino acid symbol reflects this difference. In the Weblogo representation of phosphosite motifs, we highlight in color the consensus motif as well as any charged residues if they occur most frequently, that is, in the top row of the representation.

## Phosphomotif classification

Proline-directed motifs (S/T)P, assigned as minimal Cdk motif but also recognized by MAP kinase, did not include sites adhering to the full Cdk consensus motif (S/T)Px (K/R). Both were extracted before searching for the Polo motif (D/E/N)x(S/T). R(D/E/N)x(S/T) sites were not considered Polo sites and classified as Dbf2 consensus motifs, which are also recognized by PKA/B. The Dbf2 RxxS sites do not include HxRxxS nor (K/R)(K/R)xS sites, which were classified as Cbk1 and Aurora kinase sites, respectively. Five phosphosites matched both HxRxxS and (K/R)(K/R)xS motifs. Sxx(D/E) sites overlapped with 12 (D/E/N)x(S/T), 5 (R/K)(R/K)xS, and 6 RxS motifs.

## Gene ontology

Gene ontology (GO) biological process annotation was performed on http://www.geneontology.org. Fold enrichment and *P*-values were analyzed via the PANTHER overrepresentation test (Mi *et al*, 2013).

## Phosphosite abundance change analysis

$Log_2$ H/L phosphosite ratios in our timecourse experiment were reversed to $log_2$ L/H and then normalized to 100% at time 0, corresponding to the time when cells grown in the different media, containing either heavy or light amino acids, were arrested in metaphase. Note that the different media used for labeling the "heavy" reference sample and for achieving the best synchrony of the experimental culture will have caused abundance differences of certain phosphosites between the two timepoint 0 samples. As the "heavy" reference sample remained constant during the timecourse, normalization at time 0 allows us to monitor relative changes in the experimental samples. A 50% increase in phosphosite abundance, or a decrease below 50%, over at least two consecutive timepoints was required to meet classification as phosphorylated or dephosphorylated, respectively. The timepoint of phosphorylation or dephosphorylation was then assigned as the time when the threshold was first passed. Phosphosites with values between 50 and 150% were considered stable. For reanalysis of the G1 to metaphase experiment (Godfrey *et al*, 2017), L/H phosphosite ratios were normalized to 100% at timepoint 90 min. This corresponds to the time when most cells were in mitosis and when the sample of cells grown in medium containing heavy amino acids was collected. Phosphosites were considered phosphorylated if a greater than twofold increase in abundance was recorded in at least two consecutive timepoints (the dataset is available at ProteomeXchange: PXD004461).

## Data availability

The full mass spectrometry proteomics data obtained in this study have been deposited with the ProteomeXchange Consortium via the PRIDE partner repository with the dataset identifier PXD008327.

**Expanded View** for this article is available online.

## Acknowledgements

We would like to thank G. Kelly for biostatistics support, M. Godfrey, P. Thorpe, and our laboratory members for discussions and critical reading of the manuscript. This work was supported by The Francis Crick Institute, which receives its core funding from Cancer Research UK (FC001198), the UK Medical Research Council (FC001198), and the Wellcome Trust (FC001198).

## Author contributions

SAT and FU conceived the study, SAT performed most experiments and analyses, MK investigated Clb5 proteolysis-dependent Fin1 dephosphorylation, AWJ and APS performed the mass spectrometry proteomics analysis, and SAT and FU wrote the manuscript with input from all coauthors.

## Conflict of interest

The authors declare that they have no conflict of interest.

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
