## [Review Process File · The EMBO Journal]

Phosphoproteome dynamics during mitotic exit in budding yeast

Sandra A. Touati, Meghna Kataria, Andrew W. Jones, Ambrosius P. Snijders & Frank Uhlmann

Review timeline:

Submission date:	30 November 2017
Editorial Decision:	9 January 2018
Revision received:	1 March 2018
Accepted:	20 March 2018

Editor: Hartmut Vodermaier

Transaction Report:

1st Editorial Decision

9 January 2018

Thank you for submitting your manuscript on phosphoproteome dynamics during mitotic exit, and apologies for the delay in getting back to you with a decision. At this stage, we have still only received the comments from two of the three experts who had agreed to review the study. I am now forwarding you these two fairly congruent reports together with a preliminary decision, in order to avoid unnecessary further loss of time.

As you will see, both referees appreciate the potential value of your data set, and indicate that it provides various new insights that should be of interest to the field. At the same time, they however also feel that the work remains presently mostly descriptive, and that many of the further-reaching conclusions overreach the presented data and appear overstated in light of the previous literature. In addition, they also raise a number of technical points, including insufficient methodological descriptions and uncertainties about the inclusion of biological replicates.

In light of these overall very consistent opinions from both referees, I conclude that we should be able to consider an adequately revised version further as an article in our Resource section (where we can apply less stringent requirements for mechanistic understanding). This would however be contingent on satisfactory addressing of all the specific technical and presentational concerns raised by the two reviewers, as well as rewriting of the abstract/conclusions and rephrasing of the title so as to better reflect the descriptive nature of the study. I should further remind you that this preliminary decision is still subject to change should the outstanding third report bring up important additional concerns. In case we should still receive this last report, I shall forward it to you and finalize/confirm my decision.

Thank you for the opportunity to consider this work, and please do not hesitate to contact me in case you should have any questions or comments regarding this decision and the referee reports.

REFEREE REPORTS

Referee #1:

Summary:

Touati et al. describe a phosphoproteomic analysis of mitotic exit progression in budding yeast. The results suggest, surprisingly, that most phosphorylation sites are not removed during mitotic exit and that, in fact, just as many new sites are deposited during this period as are dephosphorylated. This contradicts the current dogma that mitotic exit is marked by widespread protein dephosphorylation by phosphatases as mitotic kinases are inactivated. It reveals that mitotic exit is a much more complexly regulated period of the cell cycle that must involve tight coordination between many highly specific and dynamic kinase and phosphatase activities. As such, it is a useful paper whose content and story will be of broad interest to the cell cycle field. However, I have a list of concerns and questions below that should be addressed by the authors prior to publication.

General Concerns:

1. Most of the processed data presented in the figures looks really nice. However, some questions arose in looking at data provided in the supplemental tables. One would expect the L/H ratios at time 0 to all be close to 1, since presumably an equal amount of the two cultures are mixed and they are both identically arrested at metaphase. Yet, a cursory scan of these values reveals a surprising level of variation. As just two examples, Gin4 and Chs2 have peptides whose L/H ratios vary between 0.14-1.8 and 0.15-1.6 (more than an order of magnitude difference), respectively, even though they come from the same protein and therefore should all have the same ratio. How can this be, and what does it mean? On a related note, many L/H ratio values are negative in the tables. What does this mean? How can the ratio of two measured spectral signals give a negative value? I think some more detail about the data processing procedures and about quality assurance is needed. A legend to each supplemental table describing the processing steps would be useful.
2. Related to Point 1, I don't see any evidence from the paper that reproducibility of any of the results has been assessed. It reads as if the entire experiment was performed a single time. I would love to see at least that repeat measurements of L/H ratios at time 0 from independent cultures are consistent. This would alleviate some of the concerns from Point 1. The reporting of proteomic data of this nature from a single biological replicate is highly irregular in my experience. I understand that the experiment is long, complicated and expensive, but demonstrating reproducibility is still essential.
3. In this type of experiment there are going to be many peptides that have multiple phosphorylation sites. This presents a serious challenge for analyzing the dynamics of each site. It is unclear how the authors dealt with this problem. The peptide list in the appendix table S1 seems to only consist of singly phosphorylated peptides. Yet, there are still clear examples of multiple sites existing within the same peptide. For example, the authors discuss their Ace2 results and report distinct dynamics for sites at positions 709, 713, and 714. These three sites all reside within the same tryptic peptide. How were their individual profiles measured? This is extremely challenging to do and no information at all is provided about it. It is likely a pervasive problem affecting many of the peptides in the dataset.
4. I think some of the wording in the paper is too strong. I agree with the conclusion that kinase activity level and specificity are contributing factors to phosphorylation profiles during the cell cycle. However, they can't be the only factor, as evidenced here by the majority of Cdk and Polo sites that don't follow the early mitotic exit dephosphorylation kinetics described for a minority of the sites. For example, in the abstract the authors state "...the order arises from successive inactivation of S and M phase Cdks and of the mitotic kinase Polo." It is fair to state that successive inactivation contributes to order but not to imply that it is solely responsible for the order. Another case of oversimplifying causal effects is paragraph 1 of the Introduction, "...the balance of kinase and phosphatase activities define each substrate's phosphorylation status". Other factors must contribute as well to explain, for example, the lack of uniform dynamics of all Cdk or Polo sites here.

5. On that same note, the idea of sequential kinase inactivation generating dephosphorylation order is not novel. See, for example, Jin, F. et al. (2008). PNAS 105:16177. That, and any other papers describing this phenomenon, should be cited and the results here described as in agreement.

Specific Concerns:

Figure 1

- The FACS data in figure 1A suggest a large fraction of cells have not completed cytokinesis and returned to G1 at the 45 minute timepoint, but the plots in Figure 1B give the impression that all cells have completed anaphase, disassembled their spindles and undergone cytokinesis. Can the authors address this apparent discrepancy? No information is provided in the legend or methods on what is actually observed to generate the green plot labeled "cytokinesis" in 1B. Also, no FACS method details are provided.

Figure 2

- The legend indicates that in panel F phosphoamino acid identity is indicated, but I don't see any distinction between serine and threonine in the figure.

Figure 3

- In panel B some additional info about the Icelogo motifs would be useful. I am not familiar with that algorithm. What does "% difference" on the y axis mean? Does the analysis generate a single overall view of enriched amino acids at each position or does it generate a list of motifs with different scores (like I think Weblogo does), and if so how significant are the individual motifs shown for each timepoint relative to other motifs?

- The individual examples in panels E and F add little to the paper. It is easy to cherry pick examples from large datasets like this to highlight a point of interest. However, this can give a biased view of the overall results if not presented in the appropriate light. The fact is, there are many exceptions to the trends portrayed with Chs2 and Spa2. In fact, one can see an exception in these data themselves. While Spa2 is presented to indicate that full Cdk consensus sites are dephosphorylated before minimal Cdk sites within the same protein, the opposite is actually observed in the Chs2 data but not noted (there is a full consensus site that is the slowest to be dephosphorylated). Instead, Chs2 is used only to conclude that Cdk sites are dephosphorylated before Polo sites. I bet one could pull out examples from the data where Polo sites are dephosphorylated before minimal Cdk sites. In short, the general trends should be the focus, and because of the high level of variability, the selective use of individual examples to make conclusions about general trends should be avoided or at least minimized.

Figure 4

- No information is provided on how immunoblot signals are quantified to define dephosphorylation midpoints. I assume the midpoint is determined by measuring intensity of a phosphorylation-dependent band of reduced mobility compared to a faster migrating band presumed to be dephosphorylated. I know the authors have used some of these targets in previous papers. But the method is not described or mentioned. This is important to the conclusions in this figure. In panel A I can barely distinguish two bands for Yen1, particularly in the right side immunoblot. How are the signals distinguished from each other?

- Why is Ask1 signal needed for normalization in panel B but not in panel A or Figure EV3B? And why does the timecourse progress differently in the two strains here? It raises concerns about whether we can believe the strains progress identically in the other experiments without having that control for normalization.

Figure 5

- The conclusion in the last paragraph of page 12 seems to claim a novel observation that sequential waves of phosphorylation during cell division arise from sequential windows of individual kinase activities. But this already a widely accepted and extensively documented fundamental property of the cell cycle control system, so I don't think the authors can argue that the comparisons in Figure 5 are revealing any novel insight.

Figure 6

- In panel E the authors argue that sequential phosphorylation on Polo and Dbf2 sites occurs. But

there is a Dbf2 site (red) that is phosphorylated at exactly the same time as the Polo site. In fact, the two Dbf2 sites are quite different from each other.

Figure EV1

- Panel D is erroneously labeled as EV1C on page 8, third paragraph, line 2.

Figure EV4

- The authors should provide detail about the statistical test used and the meaning of the asterisks in panel B

Other Minor Points

- Measuring phosphorylation dynamics by SDS-PAGE mobility shift:

The data here very nicely illustrates the complex phosphorylation dynamics that often exist on multiple phosphorylated proteins. Yet SDS-PAGE mobility shift is a very common method for monitoring protein phosphorylation and in most cases the sites responsible for the mobility shift are not known. The authors use SDS-PAGE mobility shift in several experiments in support of their ideas. Do they know what sites are being monitored by the mobility shifts? It would be nice at least to note the caveat of using this method given results like Ace2 that reveal very complicated phosphorylation dynamics within a single protein.

- Can't find the list of strains. Cited as Table S4 in text but no strain info is in Table S4.

- Red and orange are difficult to distinguish when used together in the same graphs - not enough contrast. In Figure 2E I'm not sure orange was actually used although the legend indicates it was.

Referee #2:

The paper employs mass spectrometry to quantify timing of changes in phosphorylation levels on thousands of peptides over a carefully timed mitosis in budding yeast. Some specific phosphopeptides (in most cases ones known to be mitotically phosphorylated from other studies) are examined in detail. This is useful as corroboration and verification of the method, and the results provide additional useful details of timing of various sites. Many novel sites are shown to be mitotically phosphorylated and timing determined, which will be valuable information for later studies.

Much of the analysis is based on peptide sequences yielding likely identity of the phosphorylating kinase. The responsible phosphatase is generally not implicated, which is understandable since we know considerably less about primary sequence determinants of phosphatases compared to kinases.

I think it is fair to say that this paper is mainly valuable as providing a database resource for future studies; the conclusions from the work as it stands are interesting but mainly descriptive. This database will be very useful in the future, since budding yeast provides a major model system for understanding mitotic control; therefore, the relatively descriptive nature of the work does not, in my opinion, present any barrier to publication. The authors present their main conclusion as something like 'most people assume that mitotic exit is all about dephosphorylation, but there are mitotic kinases that continue to phosphorylate targets all through exit'. I understand that it can seem obligatory to come up with some 'radical novel conclusion' or whatever, in any paper published these days, but I personally don't find this conclusion at all surprising as a general statement (e.g., Dbf2, Cdc15, Cbk1 a priori seem very likely to be preferentially active after Cdc20 activation)- while the empirical, descriptive specifics presented here (what sites have what level of phosphorylation when) are findings that I obviously had no prior ideas about at all. So for me, it would be preferable to frankly call the work a useful data resource for future work and leave it at that. Or if not, find some actual data citation where someone was unwise enough to say that mitotic exit is only about dephosphorylation!

There are interesting overall conclusions reached or suggested from looking at the data. I think it is

correct that the authors lean toward the conclusion that the bulk of timing regulation of phosphopeptide level is due to kinase presence/activity, rather than to differential phosphatase presence activity through time. This is not said very explicitly, and of course the authors would not in any case completely dismiss regulated phosphatases as important contributors. However, the writing in the paper is very light in consideration of the possible role of diverse phosphatases. It is unreasonable to ask for a full mass spectrometry analysis of cdc20 block-release in phosphatase mutants compared to WT, but it is interesting to think what would happen.

Specific points/questions:

The methods description is rather terse. I assume (but it is not said) that after LC, peptides are analyzed by MS/MS (i.e. not just time-of-flight). It is also not said whether the actual site of phosphorylation within a peptide (which could contain multiple S/T residues) can be unambiguously determined.

I didn't fully get the imputation of missing data - how can a data point be missed? No signal from the heavy control for the specific peptide in some timepoint? And I assume that the rules of 'deP if lower in 2 successive timepoints' and the reverse for P did not include imputed values - this should be stated if so.

The data are based on recovery of P-peptide compared to heavy-labeled peptide from cdc20-blocked cells included as an internal control in each peptide preparation. This is fine, but there are some limitations. One is that there is no way to tell stoichiometry. A phosphopeptide on an abundant protein phosphorylated to 1% would give a signal of equal strength and detectability as a phosphopeptide on 100% of a protein present at 1% the abundance. In general it makes a considerable difference what phosphorylation stoichiometry is (e.g., an inhibitory phosphorylation occurring on only 1% of target is irrelevant biologically). I understand this is a limitation that is very hard to avoid, but it should be discussed.

Related to this, the authors do consider that a phosphopeptide could increase or decrease in abundance because the carrier protein is synthesized or degraded, without differential phosphorylation through time. They control for this by asking for some large number of random peptides from the proteome, how many alter in abundance in the time course, and few do change. This is OK as a first look but does not say anything about the specific proteins under examination in the P-peptide study. What is the overlap between the P-peptide-containing proteins and the total proteins identified looking at bulk levels? It would not be surprising if mitotically phosphorylated proteins might be preferentially subject to abundance changes (e.g., APC- or SCF-mediated degradation).

Normalization to cdc20-arrested level leads to odd quantitative results for proteins that are barely phosphorylated at that timepoint (Ace2 in Fig. 2, with LH ratios of >100). Would not percent of maximum be a more even way to quantify? Related to this, I suppose that anything not phosphorylated at all in the cdc20 block is detectable by this method - please comment.

Figure 6: unclear to me. I gather that first, phospho-peptides with a match to the given core consensus sequences were collected to give a Weblogo (not very different from the core consensus, really). The Y axis of a Weblogo is usually bits of information, which is different from the cumulative probability presented (for one thing, it is standardized to amino acid abundance overall, which is not the case for cumulative probability). The quantification is labeled '% of Plk [or other kinase] consensus motifs', with a histogram with bins corresponding to times after release. This is said in the legend to be the 'distribution of phosphorylation over time'. This could be the distribution of peak times for each individual peptide; or perhaps it represents average phosphorylation levels across the class? Either way the labeling/legend is not clear.

The Table S4 provided did not contain strain information, and I couldn't find it elsewhere.

We would like to thank the two referees for their supportive and insightful critique of our manuscript. Please find below a point-by-point response how we have used this critique to perform further analyses and experiments. We hope that the reviewers will find that our revisions have improved the manuscript.

Referee 1 acknowledged that our 'story will be of broad interest to the cell cycle field'. However, a series of concerns were raised.

1. Most of the processed data presented in the figures looks really nice. However, some questions arose in looking at data provided in the supplemental tables. One would expect the L/H ratios at time 0 to all be close to 1, since presumably an equal amount of the two cultures are mixed and they are both identically arrested at metaphase. Yet, a cursory scan of these values reveals a surprising level of variation. As just two examples, Gin4 and Chs2 have peptides whose L/H ratios vary between 0.14-1.8 and 0.15-1.6 (more than an order of magnitude difference), respectively, even though they come from the same protein and therefore should all have the same ratio. How can this be, and what does it mean? On a related note, many L/H ratio values are negative in the tables. What does this mean? How can the ratio of two measured spectral signals give a negative value? I think some more detail about the data processing procedures and about quality assurance is needed. A legend to each supplemental table describing the processing steps would be useful.

We thank the reviewer for highlighting parts of our analysis that were insufficiently documented.

It is correct that we should expect the L/H ratio at time 0 to be close to 1 (i.e. 0 on the log₂ scale). In agreement with this expectation, the distribution of ratios is indeed centred at 0. There is a certain spread, which is inevitable from the technical noise of the experiment. In addition, we do expect some phosphorylation differences at time 0 that arise from the different medium treatments used at this time. The reference culture was grown in minimal complete medium for labelling with heavy amino acids, while the experimental culture was grown in rich medium to allow for greater cell cycle synchrony.

The above explains certain deviations of the L/H ratio at time 0 from 1. Nevertheless, the 'heavy' reference sample remains constant throughout the experiment. Therefore, normalisation to time 0 allows us to report on relative phosphorylation changes as a function of mitotic progression. We have added a new section "Phosphosite abundance change analysis" to the Materials and Methods, that explains this process.

Because the L/H ratios are recorded on a log₂ scale, values can be both negative and positive. This has now been made clear in the tables and their legend. A difference between e.g. 0.14 and 1.8 is thus around three-fold.

Different phosphopeptides of the same protein can behave independently with regards to their phosphorylation status at time 0. In addition, they might be differently affected by technical noise during the data acquisition. This can explain differences of L/H ratios of different phosphopeptides of the same protein.

2. Related to Point 1, I don't see any evidence from the paper that reproducibility of any of the results has been assessed. It reads as if the entire experiment was performed a single time. I would love to see at least that repeat measurements of L/H ratios at time 0 from independent cultures are consistent. This would alleviate some of the concerns from Point 1. The reporting of proteomic data of this nature from a single biological replicate is highly irregular in my experience. I understand that the experiment is long, complicated and expensive, but demonstrating reproducibility is still essential.

The reviewer raises an important point regarding the reproducibility of large phosphoproteome studies, including ours. Despite its scale and complexity, we did indeed perform a biological repeat experiment. The revised manuscript includes a new Figure EV3, in which the results of the repeat proteomics timecourse analysis are documented and compared to the results presented in the main figures. This reveals a high degree of overlap in the phosphosites that were analysed in both repeats. Furthermore, the replicate analysis confirms all the key conclusions of our study.

3. *In this type of experiment there are going to be many peptides that have multiple phosphorylation sites. This presents a serious challenge for analyzing the dynamics of each site. It is unclear how the authors dealt with this problem. The peptide list in the appendix table S1 seems to only consist of singly phosphorylated peptides. Yet, there are still clear examples of multiple sites existing within the same peptide. For example, the authors discuss their Ace2 results and report distinct dynamics for sites at positions 709, 713, and 714. These three sites all reside within the same tryptic peptide. How were their individual profiles measured? This is extremely challenging to do and no information at all is provided about it. It is likely a pervasive problem affecting many of the peptides in the dataset.*

We agree that multiply phosphorylated peptides are challenging to analyse. As described in the Materials and Methods, all MS/MS proteomics data were searched using the Andromeda algorithm in MaxQuant. This derives SILAC quantification values on three levels: protein, peptide and modification site. For our phosphoproteomic data analysis, we exclusively used values at the individual phosphosite level, sometimes combining quantifications from multiple peptides containing the same modification (this also overcomes issues with analysis of missed cleavages). Challenges arise if confident localisation and/or quantification of individual phosphosites cannot be performed on a multiply-phosphorylated peptide, and these species can be identified in the MaxQuant output tables with a `_1` (singly-phosphorylated), `_2` (doubly-phosphorylated) or `_3` (triply-phosphorylated). For every phosphosite, the localisation probability is shown in the Appendix Table S1, sheet "all data". Only phosphosites with a localisation probability >0.85 were included in our analyses. It has been suggested that a localisation probability >0.75 provides high confidence phosphosite identification (Olsen et al. 2006, *Cell* 127, p635-648), emphasising the very high confidence of phosphosite identification that entered our analysis.

4. *I think some of the wording in the paper is too strong. I agree with the conclusion that kinase activity level and specificity are contributing factors to phosphorylation profiles during the cell cycle. However, they can't be the only factor, as evidenced here by the majority of Cdk and Polo sites that don't follow the early mitotic exit dephosphorylation kinetics described for a minority of the sites. For example, in the abstract the authors state "...the order arises from successive inactivation of S and M phase Cdks and of the mitotic kinase Polo." It is fair to state that successive inactivation contributes to order but not to imply that it is solely responsible for the order. Another case of oversimplifying causal effects is paragraph 1 of the Introduction, "...the balance of kinase and phosphatase activities define each substrate's phosphorylation status". Other factors must contribute as well to explain, for example, the lack of uniform dynamics of all Cdk or Polo sites here.*

This is an important point, we agree with the reviewer that numerous levels of input affect the phosphorylation status of any given phosphorylation site. In the revised manuscript, we explicitly refer to the examples given by the reviewer and clarify at several places throughout the text that phosphosites are regulated by numerous inputs. E.g. we have changed the abstract to read "successive inactivation of S and M phase Cdks and of the mitotic kinase Polo contributes to order these dephosphorylation events".

5. *On that same note, the idea of sequential kinase inactivation generating dephosphorylation order is not novel. See, for example, Jin, F. et al. (2008). PNAS 105:16177. That, and any other papers describing this phenomenon, should be cited and the results here described as in agreement.*

Jin et al. 2008 have analysed the contribution of FEAR and MEN-activated Cdc14 phosphatase to protein dephosphorylation during mitotic exit. They have indeed considered the possibility that sequential cyclin degradation impacts on dephosphorylation timing. Note, however, that Jin et al. 2008 did NOT observe an impact of indestructible Clb5(ddb) on dephosphorylation timing of their model substrate, Sld2 (their Figure 4F). They did observe in the same figure that Clb5(ddb) caused retention of phosphorylation in a *cdc15-2* background with compromised Cdc14 activity. This is of course relevant and we accordingly cite this study in our revised manuscript

Specific Concerns:

Figure 1 - The FACS data in figure 1A suggest a large fraction of cells have not completed cytokinesis and returned to G1 at the 45 minute timepoint, but the plots in Figure 1B give the

impression that all cells have completed anaphase, disassembled their spindles and undergone cytokinesis. Can the authors address this apparent discrepancy? No information is provided in the legend or methods on what is actually observed to generate the green plot labelled "cytokinesis" in 1B. Also, no FACS method details are provided.

Apologies for this apparent discrepancy and for not having explained this better. The FACS analysis has been performed on cells with intact cell walls. These hold daughter cells together for a while after cytokinesis, until cell wall resolution following chitinase activation. In contrast, the immunofluorescence protocol includes a spheroplasting step that removes the cell wall. Consequently, cells appear separate at the time of cytokinesis. A description of the FACS and immunofluorescence protocols is now contained in the Material and Methods, as well as a definition of cytokinesis as the appearance of unbudded daughter cells following spheroplasting. We also draw attention to this difference in the revised Figure legend.

Figure 2 - The legend indicates that in panel F phosphoamino acid identity is indicated, but I don't see any distinction between serine and threonine in the figure.

Thank you for pointing out this mistake, this has been corrected.

Figure 3 - In panel B some additional info about the Icelogo motifs would be useful. I am not familiar with that algorithm. What does "% difference" on the y axis mean? Does the analysis generate a single overall view of enriched amino acids at each position or does it generate a list of motifs with different scores (like I think Weblogo does), and if so how significant are the individual motifs shown for each timepoint relative to other motifs?

The percentage difference is a metric that compares the frequency of an amino acid in our data set to its frequency in a reference dataset. The reference data set used in our analysis is the whole *Saccharomyces cerevisiae* proteome. The height of an amino-acid symbol reflects this difference. In this way, Icelogo highlights motifs that are enriched as phosphorylation/dephosphorylation targets at each time point. An explanation is now included in a new "Sequence logo" section in the Materials and Methods, that also contains a reference to the original publication that describes the algorithm.

- The individual examples in panels E and F add little to the paper. It is easy to cherry pick examples from large datasets like this to highlight a point of interest. However, this can give a biased view of the overall results if not presented in the appropriate light. The fact is, there are many exceptions to the trends portrayed with Chs2 and Spa2. In fact, one can see an exception in these data themselves. While Spa2 is presented to indicate that full Cdk consensus sites are dephosphorylated before minimal Cdk sites within the same protein, the opposite is actually observed in the Chs2 data but not noted (there is a full consensus site that is the slowest to be dephosphorylated). Instead, Chs2 is used only to conclude that Cdk sites are dephosphorylated before Polo sites. I bet one could pull out examples from the data where Polo sites are dephosphorylated before minimal Cdk sites. In short, the general trends should be the focus, and because of the high level of variability, the selective use of individual examples to make conclusions about general trends should be avoided or at least minimized.

We acknowledge that cherry picking examples can be misleading. We have addressed this in two ways. Firstly, when we report the overall trend of earlier Cdk vs. later Polo motif dephosphorylation in Figure EV3, we now present both the median as well as the interquartile range. This gives a better impression of the behaviour of an ensemble of phosphosites. We still think that it is interesting for the reader to see examples of what can happen at the individual protein level. While the choice of protein is inevitably selective, we have extended our discussion of the dephosphorylation kinetics to draw attention to the fact that exceptions to the overall trends are common and can have a number of reasons.

Figure 4 - No information is provided on how immunoblot signals are quantified to define dephosphorylation midpoints. I assume the midpoint is determined by measuring intensity of a phosphorylation-dependent band of reduced mobility compared to a faster migrating band presumed to be dephosphorylated. I know the authors have used some of these targets in previous papers. But the method is not described or mentioned. This is important to the conclusions in this

figure. In panel A I can barely distinguish two bands for Yen1, particularly in the right side immunoblot. How are the signals distinguished from each other?

- Why is Ask1 signal needed for normalization in panel B but not in panel A or Figure EV3B? And why does the timecourse progress differently in the two strains here? It raises concerns about whether we can believe the strains progress identically in the other experiments without having that control for normalization.

Mobility shifts to monitor protein phosphorylation are a widely used tool. They facilitate the phosphorylation analysis of selected proteins in their natural environment. As suggested, we use the midpoint of the mobility shift for our comparisons. This has limitations because it is not in all cases clear which particular phosphorylation event(s) on a multiply phosphorylated protein cause the mobility shift. Note, however, that our experiment is not designed to conclude on absolute dephosphorylation timing, but rather on relative changes to dephosphorylation timing between experimental conditions. This is clarified in the revised text.

The phosphoisoforms of Yen1 are challenging to resolve by gel electrophoresis. During the revisions, we have repeated the Yen1 dephosphorylation timecourse experiment. This has led to a better resolved image that is now contained in Figure 4A. In this experiment, we have also included Ask1 as an internal dephosphorylation timing control, such that the same internal control is now included in all panels.

Figure 5 - The conclusion in the last paragraph of page 12 seems to claim a novel observation that sequential waves of phosphorylation during cell division arise from sequential windows of individual kinase activities. But this already a widely accepted and extensively documented fundamental property of the cell cycle control system, so I don't think the authors can argue that the comparisons in Figure 5 are revealing any novel insight.

Figure 5 shows more than that waves of phosphorylation arise from waves of kinase activities. The surprising finding in Figure 5 is that early rising waves ebb early and late rising waves ebb later. This is not trivial and could not have been foreseen. This is now better explained in the text.

Figure 6 - In panel E the authors argue that sequential phosphorylation on Polo and Dbf2 sites occurs. But there is a Dbf2 site (red) that is phosphorylated at exactly the same time as the Polo site. In fact, the two Dbf2 sites are quite different from each other.

This relates to a point raised in Figure 3. The referee is right that multiple inputs impact on the phosphorylation timing of any phosphosite. We have amended our description of this panel to incorporate the reviewer's observations. The panel now serves as an example to show that the consensus motifs provide only one input amongst others that impact on phosphorylation timing.

Figure EV1 - Panel D is erroneously labeled as EV1C on page 8, third paragraph, line 2.

Thank you for pointing out this mistake, which has been corrected.

Figure EV4 - The authors should provide detail about the statistical test used and the meaning of the asterisks in panel B

Apologies that p-values were missing from this analysis, these have been added to the Figure Legend. A reference to the PANTHER overrepresentation test that was used for this analysis is included in the Material and Methods section that explains the analysis.

Other Minor Points

- Measuring phosphorylation dynamics by SDS-PAGE mobility shift:

The data here very nicely illustrates the complex phosphorylation dynamics that often exist on multiple phosphorylated proteins. Yet SDS-PAGE mobility shift is a very common method for monitoring protein phosphorylation and in most cases the sites responsible for the mobility shift are not known. The authors use SDS-PAGE mobility shift in several experiments in support of their ideas. Do they know what sites are being monitored by the mobility shifts? It would be nice at least

to note the caveat of using this method given results like Ace2 that reveal very complicated phosphorylation dynamics within a single protein.

We agree with the referee that mobility shifts often reflect phosphorylation of only a subset of phosphosites on a protein. Detailed mutational analysis is required to learn which phosphorylation sites contribute to a mobility shift. We performed this analysis for Fin1, which revealed that the phospho-shift, out of six phosphosites, is primarily due to serine 148 phosphorylation. Similarly, it has been established that the Ask1 phosphoshift is primarily due to serine 250 phosphorylation (Li and Elledge, 2003). We know less about what causes the Yen1 mobility shift. This caveat is explained when we discuss the experiment shown in Figure 4 in the revised manuscript.

- Can't find the list of strains. Cited as Table S4 in text but no strain info is in Table S4.

Apologies that we forgot to include the strain list in our first submission. The strain list can now be found in Appendix Table S5.

- Red and orange are difficult to distinguish when used together in the same graphs - not enough contrast. In Figure 2E I'm not sure orange was actually used although the legend indicates it was.

Thank you for pointing out this mistake. Only red was used in Figure 2E, the Figure legend has been corrected.

Referee 2 recognises the value of a comprehensive time-resolved phosphoproteomics dataset of budding yeast mitotic exit and is supportive of publication. However, the reviewer considers the main value of our work to be that of a resource. We have no objection to publishing this work in a 'Resource' format. At the same time, we would like to emphasise that the sheer extent of protein phosphorylation during mitotic exit really was unanticipated. Of course, mitotic exit kinases are known and we cite several examples of previously documented mitotic exit phosphorylation events. Despite this, in most people's minds, mitotic exit is a time of protein dephosphorylation. E.g. David Morgan's "The Cell Cycle" textbook, a most authoritative monograph, contains the following in its table of content for the late stages of the cell cycle:

The Completion of Mitosis

7-5 Control of Late Mitosis in Budding Yeast

The protein phosphatase Cdc14 is required to complete mitosis in budding yeast

7-6 Control of Anaphase Events

Dephosphorylation of Cdk targets governs anaphase spindle behavior

7-7 Control of Telophase

Dephosphorylation of Cdk substrates drives the final steps of mitosis

In this light, the finding that an equal number of phosphosites gain phosphorylation during mitotic exit, compared to those that are dephosphorylated, was surprising at least for us.

On another point, we fully agree with the reviewer that the contribution of phosphatase specificities during mitotic exit is an important area to explore. We mention phosphatases as important regulators at several places in the revised manuscript and we include a call to study the contribution of phosphatases during mitotic exit in our revised Conclusions section.

Specific points/questions:

The methods description is rather terse. I assume (but it is not said) that after LC, peptides are analyzed by MS/MS (i.e. not just time-of-flight). It is also not said whether the actual site of phosphorylation within a peptide (which could contain multiple S/T residues) can be unambiguously determined.

These are all important points. Apologies that they weren't made clear enough. All phospho-enriched samples were analysed by an LTQ-Orbitrap Velos using triplicate injections, with each run having a different MS/MS activation method: collision-induced dissociation (CID), higher energy collisional

dissociation (HCD) and multistage activation (MSA). These techniques allowed the fragmentation of the phosphopeptides and the identification and localisation of modifications. The term MS/MS has now been included in the Materials and Methods section “Sample preparation for mass spectrometry”. Phosphorylation sites were determined using the Andromeda algorithm in MaxQuant. This derives SILAC quantification values on three levels: protein, peptide and modification site. For our phosphoproteomic data analysis, we exclusively used values at the individual phosphosite level, sometimes combining quantifications from multiple peptides containing the same modification. For every phosphosite, the localisation probability is shown in the Appendix Table S1, sheet “all data”. Only phosphosites with a very high confidence localization probability (>0.85) were included in the analysis, more stringent than the generally accepted threshold of >0.75 .

I didn't fully get the imputation of missing data - how can a data point be missed? No signal from the heavy control for the specific peptide in some timepoint? And I assume that the rules of 'deP if lower in 2 successive timepoints' and the reverse for P did not include imputed values - this should be stated if so.

The reviewer touches on a challenge of performing large-scale SILAC data analyses, to identify and quantify a specific phosphopeptide in every sample. In our experiment, we are performing quantification on 10 phospho-enriched SILAC samples, hence the chance that a phosphopeptide is missed in one of the samples is high. The number of phosphosites quantified at each time point and the number of times that phosphosites have been quantified is now depicted in Figures EV1A and B. Rather than dismissing potentially biologically important phosphosites from the downstream analyses, we performed a nearest neighbour imputation using the R package DMwR. Only phosphosites present in 6 or more timepoints were imputed. At the end, fewer than 12% of all values were imputed in the mitotic exit dataset. Further details on the imputation can now be found in the Materials and Methods section “Filtration, imputation and smoothing”. The same method was used to curate a previous large-scale SILAC dataset (Swaffer et al. 2016, *Cell* 167, p1750-1761).

The data are based on recovery of P-peptide compared to heavy-labeled peptide from cdc20-blocked cells included as an internal control in each peptide preparation. This is fine, but there are some limitations. One is that there is no way to tell stoichiometry. A phosphopeptide on an abundant protein phosphorylated to 1% would give a signal of equal strength and detectability as a phosphopeptide on 100% of a protein present at 1% the abundance. In general, it makes a considerable difference what phosphorylation stoichiometry is (e.g., an inhibitory phosphorylation occurring on only 1% of target is irrelevant biologically). I understand this is a limitation that is very hard to avoid, but it should be discussed.

This is a really good point, the analysis of stoichiometry and occupancy would be beneficial and very interesting. However, this is a major technical task. The calculation of occupancy requires the identification and quantification of the same tryptic peptides in both phospho-enriched and non-enriched samples (see e.g. Olsen et al. 2006, *Cell* 127, p635-648). Our study focussed on phosphopeptide enrichment, such as to identify as many as possible phosphosites. We would not be able to identify the same number of peptides in their unmodified form. Olsen et al. were able to determine the occupancy of approximately 5% of detected phosphosites. In our study, we detected unmodified peptides corresponding to 6.2% of the phosphosites. This should allow us to quantify phosphorylation occupancy on this subset, which poses its own challenges, in the future. We raise this important future direction in our revised Conclusions section.

Related to this, the authors do consider that a phosphopeptide could increase or decrease in abundance because the carrier protein is synthesized or degraded, without differential phosphorylation through time. They control for this by asking for some large number of random peptides from the proteome, how many alter in abundance in the time course, and few do change. This is OK as a first look but does not say anything about the specific proteins under examination in the P-peptide study. What is the overlap between the P-peptide-containing proteins and the total proteins identified looking at bulk levels? It would not be surprising if mitotically phosphorylated proteins might be preferentially subject to abundance changes (e.g., APC- or SCF-mediated degradation).

This is another very good point. We have now included in Figure EV1D a Venn diagram showing the overlap of the proteins identified in the phospho-enriched and non-enriched datasets. There are 362 proteins in common. We also provide a table that documents the behaviour of those proteins

that undergo phosphorylation changes. E.g. of the 59 common proteins found to be 'dephosphorylated' on one or more sites, only 1 protein decreased in abundance. This suggests that protein degradation occurs, but is not a major confounding factor in our phosphosite analysis.

Normalization to cdc20-arrested level leads to odd quantitative results for proteins that are barely phosphorylated at that timepoint (Ace2 in Fig. 2, with LH ratios of >100). Would not percent of maximum be a more even way to quantify? Related to this, I suppose that anything not phosphorylated at all in the cdc20 block is detectable by this method - please comment.

We chose metaphase as the reference point because the heavy SILAC sample was taken at this time, thus providing the most meaningful normalisation. Our downstream analysis in turn is facilitated by a single reference point. We did try to define an alternative reference point. However, those phosphosites that increase in abundance reach their maximum at various points, in frequent cases losing phosphorylation again at later times. This made the definition of an alternate reference point difficult.

Because of the high dynamic range, phosphosites can typically be detected and an L/H ratio calculated at every time point, even if signal levels are low. Both, phosphosites that are low in metaphase, as well as those that are dephosphorylated during mitotic exit, remain detectable at a low level at their extreme time points.

Figure 6: unclear to me. I gather that first, phospho-peptides with a match to the given core consensus sequences were collected to give a Weblogo (not very different from the core consensus, really). The Y axis of a Weblogo is usually bits of information, which is different from the cumulative probability presented (for one thing, it is standardized to amino acid abundance overall, which is not the case for cumulative probability). The quantification is labeled '% of Plk [or other kinase] consensus motifs', with a histogram with bins corresponding to times after release. This is said in the legend to be the 'distribution of phosphorylation over time'. This could be the distribution of peak times for each individual peptide; or perhaps it represents average phosphorylation levels across the class? Either way the labeling/legend is not clear.

We apologise that this was unclear. We first discovered enriched phosphosite motifs using the enrichment analysis provided by IceLogo. After that, we used Weblogo to display the surrounding regions of all the phosphosites that adhere to a discovered motif. This was done to explore the motif surroundings. In several cases, the Weblogo representation highlighted additional charged regions outside the core consensus motifs. This is better explained in the revised manuscript.

The graphs beneath the Weblogos show the distribution of (de)phosphorylation times, i.e. the times when phosphosites that adhere to a particular motif pass the 50% or 150% threshold. For more clarity, we have relabelled those graphs and improved the Figure legends.

The Table S4 provided did not contain strain information, and I couldn't find it elsewhere.

Apologies that we forgot to include the strain list in our first submission. The strain list can now be found in Appendix Table S5.

2nd Editorial Decision

20 March 2018

Thank you for submitting your revised manuscript for our consideration. Referee 1 has now looked at it once more, and as you will see from their comments copied below, has no further objections towards acceptance and publication in The EMBO Journal.

 REFEREE REPORT

Referee #1 (Report for Author)

The authors have thoughtfully and thoroughly addressed all concerns with the original manuscript.

They have performed an additional replicate of the entire proteomics experiment to demonstrate reproducibility of the general results and have added many additional explanations to the text that greatly improve the clarity.

The one point that I still have some problems with is the issue of multi-site phosphorylation and site localization. I understand the scoring algorithm from Maxquant, but just looking at the Ace2 sites because they are first in the list: the sites at 709, 713 and 714 are all within the same tryptic segment. Each one is apparently identified with confidence based on the site score on a unique peptide sequence with either 5 or 6 missed cleavages. I just have a hard time believing that these unusual peptides accurately reflect distinct kinetics (one is reported as dephosphorylated while the other two are reported as phosphorylated). This is a general problem with phosphoproteomic data interpretation though, and not something the authors have done incorrectly. The general conclusions of the paper are sound and I recommend publication at this point.

Corresponding Author Name: Frank Uhlmann

Journal Submitted to: The EMBO Journal

Manuscript Number: EMBOJ-2017-98745